# On the Importance of Pretraining Data Alignment for Atomic Property Prediction

**Yasir Ghunaim**                                                       *yasir.ghunaim@kaust.edu.sa*
*King Abdullah University of Science and Technology (KAUST)*

**Hasan Abed Al Kader Hammoud**                    *hasanabedalkader.hammoud@kaust.edu.sa*
*King Abdullah University of Science and Technology (KAUST)*

**Bernard Ghanem**                                                   *bernard.ghanem@kaust.edu.sa*
*King Abdullah University of Science and Technology (KAUST)*

**Reviewed on OpenReview:** *https://openreview.net/forum?id=jfD9BsrDTb*

## Abstract

This paper challenges the recent paradigm in atomic property prediction that links progress to growing dataset sizes and computational resources. We show that pretraining on a carefully selected task-aligned dataset can match or even surpass large-scale joint pretraining while using only 1/24th of the pretraining budget. We introduce the Chemical Similarity Index (CSI), a simple metric for molecular graphs inspired by the Fréchet Inception Distance in computer vision, which quantifies the alignment between upstream pretraining datasets and downstream tasks. By selecting the most aligned dataset with minimal CSI distance, we show that models pretrained on a smaller, focused dataset consistently achieve better performance on downstream tasks than those pretrained on massive, mixed datasets such as JMP. This holds even when the mixed dataset includes the upstream dataset most aligned with the downstream task. Counterintuitively, we also find that indiscriminately adding more data can degrade model performance when the additional data is poorly aligned with the target task. Our findings highlight that **quality often outperforms quantity** in pretraining for atomic property prediction. The code is publicly available at: github.com/Yasir-Ghunaim/efficient-atom.

## 1 Introduction

Machine learning is transforming molecular modeling, driving advancements in accurate predictions and simulations of molecular behavior (Chanussot et al., 2021; Tran et al., 2023; Liao et al., 2023). These breakthroughs directly impact the acceleration of progress in crucial fields such as drug discovery (Huang et al., 2021) and global climate change mitigation (Sriram et al., 2024). The improvements in this field have been primarily attributed to innovations in model architectures (Liao et al., 2023; Gasteiger et al., 2021; Passaro & Zitnick, 2023) and the growing availability of large-scale molecular datasets. In recent years, the sizes of molecular datasets have increased dramatically, from tens of thousands of examples (Christensen & Von Lilienfeld, 2020; Chmiela et al., 2023; Wu et al., 2018) to hundreds of millions (Chanussot et al., 2021; Tran et al., 2023). This rapid growth in scale has also caused a surge in the computational resources required for pretraining, increasing from a few days on a single GPU to over a thousand GPU-days (Shoghi et al., 2023; Liao et al., 2023). This trend raises the question:

💡 *Is scaling data and resources the only path forward for pretraining in atomic property prediction, or can intelligent data selection achieve similar performance more efficiently?*

While data selection strategies for pretraining have been explored in fields like natural language processing (Penedo et al., 2024) and computer vision (Hammoud et al., 2024; Li et al., 2023), this area remains

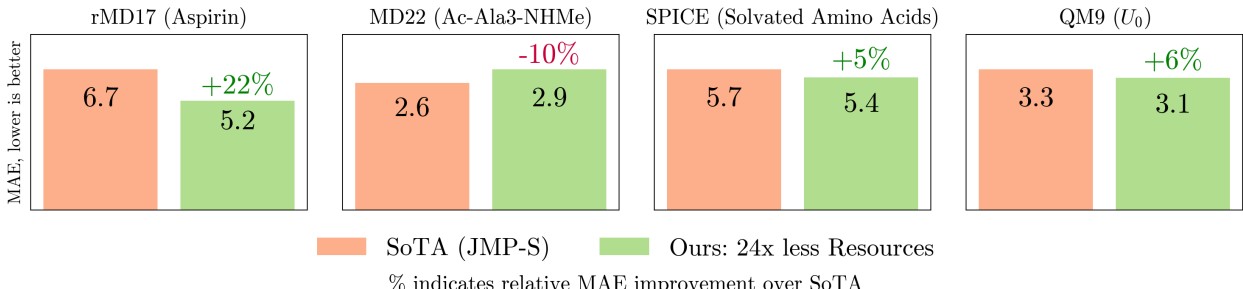

Figure 1: **Pretraining on a High-Quality, Task-Aligned Dataset.** Pretraining on a carefully selected high-quality dataset achieves comparable or superior mean absolute error (MAE) across tasks while reducing pretraining budget by a factor of 24 compared to JMP-S, which is pretrained on all upstream datasets. Lower MAE indicates better performance.

largely underexplored in atomic property prediction, where unique challenges arise. This gap is especially critical for 3D molecular and material pretraining, which requires far more computation (Liao et al., 2023) than methods based on 2D representations (Rong et al., 2020). In this paper, we challenge the prevailing assumption that "bigger is better" by exploring whether a smaller, strategically selected dataset can lead to comparable or even superior performance while substantially reducing computational demands. We introduce a pretraining paradigm for 3D molecular and material systems that shifts the focus from data and compute scaling to selecting the most relevant upstream dataset for improved downstream performance.

Our experiments reveal two key insights: **(1) Competitive Performance Can Be Achieved with 24× Fewer Resources:** Selecting upstream datasets based on downstream alignment matches or exceeds large-scale pretrained models such as JMP (Shoghi et al., 2023), while using only 10M training instances instead of JMP's 240M-instance pretraining budget (a 24× reduction under our budget definition), as shown in Figure 1. **(2) Quality Outperforms Quantity:** Expanding the pretraining dataset by incorporating additional data from less aligned sources can harm downstream performance rather than improve it.

To explore the potential of dataset selection for pretraining in atomic property prediction, we introduce the Chemical Similarity Index (CSI), a simple metric inspired by the Fréchet Inception Distance (FID) from computer vision. CSI measures the alignment between an upstream dataset and a downstream task, enabling the selection of chemically relevant pretraining data. By focusing on these highly relevant datasets, we significantly reduce computational costs while maintaining competitive performance and, in many cases, achieving improvements. While large-scale datasets like OC20 (Chanussot et al., 2021; Tran et al., 2023) and mixed datasets like JMP (Shoghi et al., 2023) are popular choices for pretraining in molecular domains (Kolluru et al., 2022; Shoghi et al., 2023), our findings challenge their universal utility. Surprisingly, pretraining on a single, carefully selected dataset guided by CSI often outperforms models trained on mixtures, even when those include the most relevant dataset.

**The contributions of this paper are threefold:** (1) We introduce a novel framework for computationally efficient pretraining of molecular machine learning models, demonstrating that strategic data selection can match or outperform models trained on much larger datasets. (2) We propose the Chemical Similarity Index (CSI), a metric for assessing the alignment between upstream and downstream molecular datasets, enabling effective dataset selection. (3) We provide an extensive empirical evaluation demonstrating the effectiveness of our approach, offering a practical and efficient alternative to the current trend of ever-increasing data and computational costs in molecular machine learning.

## 2 Related Work

**Pretraining for Atomic Property Prediction.** Inspired by the success of pretraining in computer vision and natural language processing, pretraining for atomic property prediction has gained significant attention in recent years. Most approaches in molecular machine learning have focused on self-supervised pretraining (Rong et al., 2020; Liu et al., 2021; Jiao et al., 2023; Chen et al., 2021; Zhou et al., 2022a;

Ji et al., 2024), while fewer studies have explored the effectiveness of supervised pretraining (Smith et al., 2019; 2018; Kolluru et al., 2022). Early self-supervised methods such as GROVER (Rong et al., 2020) target 2D molecular graphs. More recent approaches, including GraphMVP (Liu et al., 2021), Uni-Mol (Zhou et al., 2022a) and Uni-Mol2 (Ji et al., 2024), extend to 3D structures and often use denoising objectives that are best suited for equilibrium geometries (Liao et al., 2024). This limits their applicability to large-scale non-equilibrium datasets common in molecular and materials modeling. In contrast, recent supervised pretraining frameworks such as Joint Multi-domain Pre-training (JMP) (Shoghi et al., 2023) jointly pretrain on diverse large-scale labeled datasets and operate effectively on non-equilibrium data. While powerful, JMP requires substantial compute due to the massive data volume and does not reveal how each pretraining dataset influences downstream performance. Our work addresses this gap by systematically analyzing the link between pretraining dataset choice and downstream performance. Based on our analysis, we show that selecting a single task-aligned upstream dataset can match or exceed joint pretraining, enabling efficient model development under constrained computational budgets.

**Computational Budgeting.** Recent research highlights the importance of studying model performance under computationally budgeted setups. In continual learning (CL), works by Prabhu et al. (2023) and Ghunaim et al. (2023) show that simple baselines often outperform state-of-the-art methods in compute-constrained settings. TiC-CLIP (Garg et al., 2024) further demonstrates efficient rehearsal-based training for time-continuous data. For Vision Transformers, Pan et al. (2022) propose a framework to dynamically control model complexity during training, achieving competitive performance under varying budgets. Li et al. (2019) formalize budgeted training, showing that budget-aware learning rate schedules, such as linear decay, are critical for robust performance across tasks like image classification and object detection. In multi-domain learning, Berriel et al. (2019) introduce Budget-Aware Adapters, which reduce computational complexity while maintaining accuracy by selecting relevant feature channels. These findings across domains emphasize the critical need for more efficient approaches that can achieve competitive performance while minimizing computational costs.

**Distribution Similarity.** A variety of methods exist for quantifying the distance between probability distributions, including Kullback–Leibler (KL) divergence (Kullback & Leibler, 1951), Jensen–Shannon (JS) divergence (Lin, 2002), Maximum Mean Discrepancy (MMD) (Gretton et al., 2012), and the Fréchet Inception Distance (FID) (Heusel et al., 2017). FID has become a standard metric in computer vision for comparing real and generated data distributions by evaluating differences in the means and covariances of their feature representations. In the chemical domain, the Fréchet ChemNet Distance (FCD) (Preuer et al., 2018) adapts FID for evaluating generative models that use SMILES-based molecular representations and has been integrated into benchmarking frameworks such as GuacaMol (Brown et al., 2019). While FCD demonstrates the applicability of FID-based metrics in chemistry, it is primarily applied to SMILES-based molecular data within a single chemical domain. Our proposed Chemical Similarity Index (CSI) is also derived from FID, but is designed for 3D graph-structured systems and focuses on measuring alignment across multiple domains such as molecules and materials. To our knowledge, we are the first to systematically study pretraining dataset alignment in 3D atomic property prediction.

**Data Selection.** Efficient training through data selection has been explored via two primary approaches: subset selection and dataset distillation. Subset selection aims to identify a representative subset of the training data that matches or even outperforms training on the full dataset. Several methods have been proposed for vision and NLP tasks (Attendu & Corbeil, 2023; Killamsetty et al., 2021a;b; Kaushal et al., 2019; Bairi et al., 2015; Lapedriza et al., 2013). Dataset distillation, introduced by Wang et al. (2018), focuses on generating a smaller, synthetic subset of the dataset that preserves performance while reducing training time and storage requirements. Subsequent work has explored techniques such as meta-learning (Zhou et al., 2022b; Nguyen et al., 2021a;b), gradient matching (Zhao et al., 2021), and distribution matching (Zhao & Bilen, 2023). While most research in distillation has focused on vision tasks, a few studies have extended it to graph data (Jin et al., 2022b; Liu et al., 2022; Jin et al., 2022a), though primarily targeting knowledge and social graphs rather than molecular graphs.

Two recent vision studies are particularly relevant to our work. First, Hammoud et al. (2024) shows that increasing pretraining data diversity enhances performance only when distribution shifts between upstream and downstream tasks are minimized. Second, Li et al. (2023) introduces a method to dynamically leverage

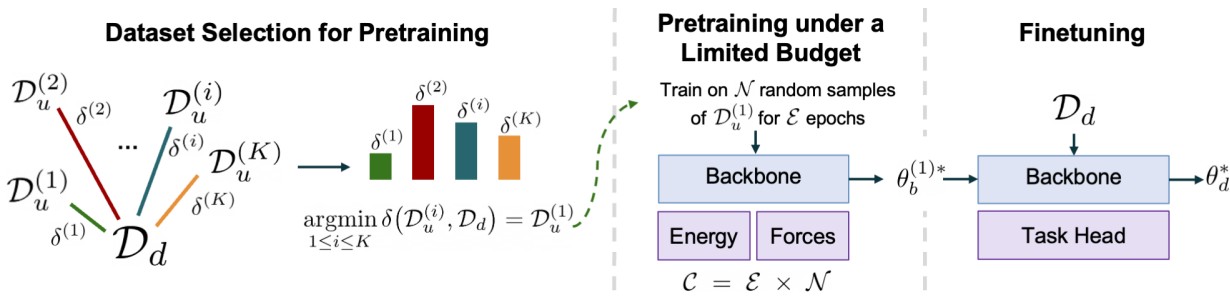

Figure 2: **Pipeline Overview**. Our paradigm for pretraining and finetuning consists of two new components: (1) *Dataset Selection Stage*, where a distance metric $\delta$ is employed to identify the dataset that is most similar to our downstream task dataset $\mathcal{D}_d$, in this case $\mathcal{D}_u^{(1)}$. This selected dataset is then used for pretraining the model. (2) *Limited Budget Pretraining*, where we impose a training budget by subsampling $\mathcal{N}$ random samples from $\mathcal{D}_u^{(1)}$ and training the model for $\mathcal{E}$ epochs. This results in a computational budget of $\mathcal{C} = \mathcal{E} \times \mathcal{N}$. The pretrained backbone $\theta_b^{(1)*}$ is subsequently finetuned on the downstream task dataset $\mathcal{D}_d$ to obtain the final model parameters $\theta_d^*$.

the open web, reducing the distribution gap between upstream and downstream tasks through targeted representation learning. Findings from other domains suggest that aligning upstream datasets may be crucial for effective pretraining.

**Comparison to Our Work.** To the best of our knowledge, no prior work has specifically explored upstream dataset selection for molecular graphs, which present unique challenges due to their structural and chemical complexity. In this work, we take the first step in addressing this gap by focusing on aligning upstream and downstream distributions at the dataset level rather than subselecting at a sample-wise level or creating a synthetic distilled version of the dataset.

## 3 Formulation and Setup

In this section, we present our problem setup, notion of a computational budget, and the formulation of dataset similarity. We then detail how we adapt the Fréchet Inception Distance (FID) to the molecular domain, yielding the *Chemical Similarity Index (CSI)*. Our setup is illustrated in Figure 2. Throughout this work, we use the term 'molecular' broadly to encompass both molecular and materials domains, as well as their respective datasets.

### 3.1 Formal Setting

**Upstream and Downstream Datasets.** Let $\{\mathcal{D}_u^{(1)}, \mathcal{D}_u^{(2)}, \ldots, \mathcal{D}_u^{(K)}\}$ denote a collection of $K$ *upstream* (pretraining) datasets, each containing molecular structures paired with relevant atomic properties (e.g., energies and forces). In the typical paradigm, upstream datasets are typically aggregated into a single pretraining set:

$$\mathcal{D}_u = \bigcup_{i=1}^{K} \mathcal{D}_u^{(i)}. \tag{1}$$

We further define $\mathcal{D}_d$ as the *downstream* dataset, which focuses on a specific prediction task (e.g., predicting an atomic property).

**Multi-task Pretraining.** We consider a neural network $\Phi(\cdot; \theta)$, where $\theta$ encompasses the shared backbone parameters $\theta_b$ and task-specific head parameters $\theta_e$ (for energy prediction) and $\theta_f$ (for force prediction). During pretraining, the network is trained to simultaneously predict energies and forces. Formally, the multi-task pretraining objective over an upstream dataset $\mathcal{D}_u^{(i)}$ is given by:

$$\theta^{(i)*} = \arg\min_{\theta} \mathcal{L}_{\text{pretrain}}(\theta; \mathcal{D}_u^{(i)}), \tag{2}$$

where $\theta = \{\theta_b, \theta_e, \theta_f\}$ and

$$\mathcal{L}_{\text{pretrain}}(\theta; \mathcal{D}_u^{(i)}) \;=\; \alpha \, \mathcal{L}_{\text{energy}}(\theta_b, \theta_e; \mathcal{D}_u^{(i)}) \;+\; \beta \, \mathcal{L}_{\text{forces}}(\theta_b, \theta_f; \mathcal{D}_u^{(i)}). \tag{3}$$

We compute $\mathcal{L}_{\text{energy}}$ using the Mean Absolute Error (MAE) and $\mathcal{L}_{\text{forces}}$ using the mean per-atom Euclidean (L2) distance. Coefficients $\alpha$ and $\beta$ weight the importance of energy and force tasks, respectively. Following the JMP paper (Shoghi et al., 2023), we set $\beta > \alpha$ to prioritize accurate force predictions in atomistic modeling. Pretraining can be performed on either the joint upstream dataset $\mathcal{D}_u$, similar to JMP (Shoghi et al., 2023), or on an individual upstream dataset $\mathcal{D}_u^{(i)}$, as in our selective setting.

**Fine-Tuning.** After the multi-task pretraining phase, the task-specific heads $\theta_e$ and $\theta_f$ are discarded, and a new task-specific head $\theta_h$ is attached to the pretrained backbone $\theta_b$. The downstream objective then becomes:

$$\theta_d^* \;=\; \arg\min_{\theta_b, \theta_h} \; \mathcal{L}_{\text{finetune}}(\theta_b, \theta_h; \theta_b^{(i)*}, \mathcal{D}_d), \tag{4}$$

where $\theta_b^{(i)*}$ denotes the pretrained backbone parameters from Eq. (2). Intuitively, the downstream training refines the shared backbone parameters $\theta_b$ and learns the task-specific head $\theta_h$ to capture the target property in $\mathcal{D}_d$.

*In this paper, we introduce two additional definitions for this setting: (1) computational budget and (2) dataset similarity.*

**Computational Budget.** Following Hammoud et al. (2024), we define the *computational budget* $\mathcal{C}$ to be the product of the number of epochs $\mathcal{E}$ and the number of unique samples $\mathcal{N}$ in the pretraining dataset:

$$\mathcal{C} \;=\; \mathcal{E} \;\times\; \mathcal{N}. \tag{5}$$

Hence, the computational budget $\mathcal{C}$ represents the total number of samples processed over training. It naturally splits into two factors: the dataset size ($\mathcal{N}$) and the number of passes through it ($\mathcal{E}$). The choice of $\mathcal{C}$ depends on the available computing resources. In our main experiments, we fix $\mathcal{C}$, $\mathcal{N}$, and $\mathcal{E}$ to ensure a fair comparison across different upstream datasets. We also include experiments in which $\mathcal{N}$ (and thus $\mathcal{C}$) varies, in order to analyze the impact of dataset size and total compute on downstream performance.

**Dataset Similarity.** A key objective of this work is to estimate how well an upstream dataset $\mathcal{D}_u$ aligns with a downstream dataset $\mathcal{D}_d$. We therefore seek a distance metric

$$\delta(\mathcal{D}_u, \mathcal{D}_d)$$

that quantifies their alignment or "similarity." In principle, a lower value of $\delta(\mathcal{D}_u, \mathcal{D}_d)$ reflects a higher degree of alignment between the upstream and downstream distributions. Thus, among multiple candidate upstream datasets $\{\mathcal{D}_u^{(1)}, \ldots, \mathcal{D}_u^{(K)}\}$, the one that minimizes

$$\underset{1 \leq i \leq K}{\arg\min} \, \delta\big(\mathcal{D}_u^{(i)}, \mathcal{D}_d\big)$$

should provide the most effective pretraining for $\mathcal{D}_d$. In this paper, we empirically test this assumption, examining whether lower $\delta$-values indeed correlate with improved downstream performance. Motivated by this, we use $\delta$ as a principled metric to guide dataset selection for Eq. (1). Instead of aggregating all upstream datasets, we modify the pretraining setup to use only the single dataset $\mathcal{D}_u^{(i)}$ that best aligns with the downstream task under a fixed computational budget.

## 3.2   The Chemical Similarity Index (CSI)

**Recap of FID.** Our proposed *Chemical Similarity Index (CSI)* draws its inspiration from the well-known Fréchet Inception Distance (FID) (Heusel et al., 2017). FID is commonly used in computer vision to compare two sets of images via their feature distributions. Specifically, if one extracts features (e.g., from an Inception

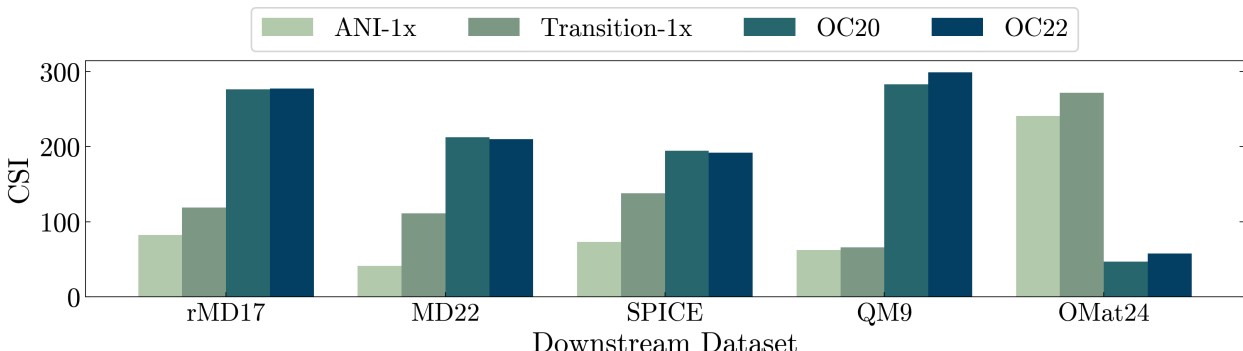

Figure 3: **Alignment Between Upstream and Downstream Using CSI.** We assess how well the extracted representations from each upstream dataset align with downstream tasks using our CSI metric, where lower values indicate stronger alignment. ANI-1x demonstrates the closest feature alignment with most downstream tasks. However, for OMat24, the catalysis datasets OC20 and OC22 exhibit the strongest alignment.

network) for datasets $X$ and $Y$ and denotes their empirical means and covariances by $\mu_X, \Sigma_X$ and $\mu_Y, \Sigma_Y$, then

$$\text{FID}(X, Y) \ = \ \|\mu_X - \mu_Y\|^2 \ + \ \text{Tr}\Big(\Sigma_X + \Sigma_Y - 2(\Sigma_X \Sigma_Y)^{1/2}\Big). \tag{6}$$

The central idea is to represent each sample in a feature space where distances encode semantic similarity and then compare the distributions of these representations for the two datasets.

To adapt FID for graph-structured molecular data, we compute the CSI metric using node embeddings as features and apply class-balanced sampling to ensure representative coverage of molecular types in each upstream dataset. For computational feasibility, we subsample 10k instances from both the upstream and downstream datasets. To keep the metric fixed and comparable across all evaluations, we extract features using EquiformerV2 (Liao et al., 2023) pretrained on OC20 (Chanussot et al., 2021). We provide results with additional pretrained checkpoints in Appendix B, feature representation choices in Appendix C, and class-balancing procedure in Appendix D.

**CSI Between Upstream and Downstream Results.** In Figure 3, we present the CSI values for pairs of upstream and downstream tasks related to energy and force predictions, with additional details about the datasets and targets provided in Section 4. For the first four downstream datasets in Figure 3, ANI-1x (Smith et al., 2020) consistently achieves the closest alignment, reflecting its design goal of maximizing chemical diversity. Transition-1x (Schreiner et al., 2022), which focuses on transition states, ranks second suggesting that its emphasis on high-energy transition states leads to partial overlap with downstream distributions. OMat24 (Barroso-Luque et al., 2024), which contains inorganic materials, is more closely aligned with the catalysis datasets OC20 (Chanussot et al., 2021) and OC22 (Tran et al., 2023) than with other upstream datasets. While OC20 and OC22 are often favored for pretraining (Shoghi et al., 2023; Kolluru et al., 2022) due to their scale and chemical diversity, our metric suggests they may not align well with most of the considered downstream tasks. Next, we examine whether these alignment values correlate with downstream performance.

## 4 Experiments

We evaluate the impact of pretraining on different upstream datasets for downstream performance and investigate how well the CSI values in Figure 3 reflect the relevance of these datasets. We begin by defining the datasets, baselines, and evaluation setup.

**Upstream Datasets:** Following JMP (Shoghi et al., 2023), we perform pretraining on upstream datasets of small molecules, including ANI-1x (Smith et al., 2020) and Transition-1x (Schreiner et al., 2022), as well as large-scale catalysis datasets, OC20 (Chanussot et al., 2021) and OC22 (Tran et al., 2023). These datasets

vary in domain focus and graph size, enabling us to examine how these factors impact the generalization of pretraining across downstream tasks. The ground-truth labels, energy and forces, are computed using Density Functional Theory (DFT).

**Downstream Datasets:** For downstream evaluation, we focus on in-distribution (ID) tasks involving energy or force prediction, following the definition in JMP (Shoghi et al., 2023). We discuss out-of-distribution (OOD) tasks in Section 5. To keep the experiments tractable, we evaluate on a single target per downstream dataset, since we run six baselines for each. These targets and their corresponding datasets are: Aspirin in rMD17 (Christensen & Von Lilienfeld, 2020), Ac-Ala3-NHMe in MD22 (Chmiela et al., 2023), solvated amino acids in SPICE (Eastman et al., 2023), $U_0$ in QM9 (Wu et al., 2018), and force prediction on the rattled-300-subsampled split of OMat24 (Barroso-Luque et al., 2024).

**Baselines:** We report the original performance of JMP, where "JMP-S" and "JMP-L" correspond to the small and large backbones, respectively. Additionally, we present our reproduced fine-tuning results using the official JMP checkpoints, denoted as "JMP-S$^*$" and "JMP-L$^*$".

For our budgeted evaluation, we present results in two categories: pretraining on a single upstream dataset and pretraining on a joint combination of all upstream datasets. For single-dataset experiments, we randomly sample $\mathcal{N}$ instances from the original upstream data. For joint pretraining, we construct the training set using two different strategies. (1) Balanced Sampling, where an equal number of samples is drawn from each of the four upstream datasets, totaling $\mathcal{N}$ samples; and (2) Temperature-Based Sampling, which preserves the dataset proportions used in the full 120M sample set of JMP (Shoghi et al., 2023).

**Evaluation Setup:** We pretrain the GemNet-OC-S model (Gasteiger et al., 2022) on each individual upstream dataset, as well as on joint configurations that combine all upstream datasets, following the baseline setups. For our main experiments, we set a fixed computational budget of $\mathcal{C} = 10M$, achieved by training on $\mathcal{N} = 2M$ samples for $\mathcal{E} = 5$ epochs. This represents a 24× reduction in pretraining budget compared to that used in JMP (Shoghi et al., 2023). Our budget ensures accessibility and reproducibility, with each pretraining run completing within 1 to 2 days on an A100 GPU. Additional budget configurations are explored in later sections and the appendix. Each pretrained model is then fine-tuned separately on each downstream task.

### 4.1 Does CSI Correlate with Better Performance?

In Figure 3, we presented CSI values quantifying the alignment between each upstream and downstream dataset. These results raise a critical question:

💡 *Can CSI reliably guide the selection of pretraining datasets to achieve optimal performance on specific downstream tasks?*

Table 1 summarizes the downstream performance of models pretrained on different datasets in the in-distribution setting. Consistent with the CSI values, the upstream dataset with the lowest CSI for each downstream task also achieves the best fine-tuning performance. Specifically, pretraining on ANI-1x, despite being the smallest dataset, yields the best fine-tuning results on rMD17, MD22, SPICE and QM9, outperforming all other individual pretraining datasets as well as the joint variants. For instance, on the rMD17, SPICE and QM9 datasets, the model pretrained on ANI-1x achieves MAEs of 5.2, 5.39 and 3.1, compared to 6.7, 5.71 and 3.3 for JMP-S. This strong performance is achieved with less than 5% of the pretraining budget used by JMP-S. For OMat24, the model pretrained on OC20 achieves the best performance among the limited-budget baselines with an MAE of 87.4, comparable to JMP-S's 84.4 despite its much larger 240M pretraining budget. This outcome is also predicted by CSI, which ranks OC20 as the most similar upstream dataset for this task.

Furthermore, joint pretraining, whether balanced or temperature-based, generally performs worse than individual pretraining on the most CSI-aligned dataset for each task when both are trained under the same limited-budget setting. Following the JMP formulation, temperature-based sampling allocates more samples to OC20 and OC22 than balanced sampling. On OMat24, where OC20 and OC22 are most aligned according to CSI values, temperature-based sampling outperforms balanced sampling. However, temperature-based sampling degrades performance on other tasks where OC20 and OC22 are less aligned with the downstream

Table 1: **In-Distribution Evaluation for Energy and Force Targets.** We report test MAE when fine-tuning on downstream targets, as detailed in Downstream Datasets (Section 4). The top section represents models pretrained with the large-scale JMP budget, while the lower two sections show results under a limited budget. JMP-S* denotes reproduced results. We report the average and standard deviation over three random seeds.

| $\mathcal{C}$ | Upstream Data | Backbone | rMD17 (meV/Å) | MD22 (meV/Å) | SPICE (meV/Å) | QM9 (meV) | OMat24 (meV/Å) |
|---|---|---|---|---|---|---|---|
| 240M | Joint (Temperature) | JMP-L (GemNet-OC-L) | 5.1 | 1.92 | 4.75 | 2.9 | - |
| | | JMP-S (GemNet-OC-S) | 6.7 | 2.64 | 5.71 | 3.3 | - |
| | | JMP-S*(GemNet-OC-S) | 6.8 | 3.21 | 5.60 | 3.4 | 84.4 |
| 10M | ANI-1x | GemNet-OC-S | **5.2 ± 0.2** | **2.92 ± 0.03** | **5.39 ± 0.26** | **3.1 ± 0.1** | 98.1 ± 0.5 |
| | Transition-1x | | 9.8 ± 0.5 | 3.70 ± 0.02 | 7.70 ± 0.31 | 3.6 ± 0.3 | 100.8 ± 0.1 |
| | OC20 | | 13.8 ± 0.7 | 5.10 ± 0.57 | 9.67 ± 0.73 | 5.3 ± 0.4 | **87.4 ± 0.1** |
| | OC22 | | 15.3 ± 0.7 | 5.34 ± 0.18 | 10.62 ± 0.42 | 5.7 ± 0.1 | 92.5 ± 0.1 |
| 10M | Joint (Balanced) | GemNet-OC-S | 9.1 ± 1.8 | 3.73 ± 0.13 | 6.99 ± 0.33 | 3.5 ± 0.2 | 90.3 ± 0.1 |
| | Joint (Temperature) | | 11.0 ± 1.5 | 4.41 ± 0.59 | 8.44 ± 0.63 | 4.0 ± 0.4 | 88.4 ± 0.1 |

dataset. These results suggest that, under a limited budget, mixing upstream datasets with varying CSI values is suboptimal and requires significantly larger pretraining budgets to achieve competitive performance.

**Takeaway.** Our experiments reveal three key insights for in-distribution downstream tasks: (1) Task-aligned upstream datasets outperform larger joint datasets. (2) Joint pretraining can match the benefits of pretraining on a highly task-aligned dataset, but doing so requires substantially larger pretraining budgets. (3) CSI effectively predicts downstream performance, as lower CSI values consistently correlate with better results.

## 4.2 What is the Effect of Computational Budget?

Building on our earlier findings, we now investigate how varying the computational budget impacts downstream performance. Specifically, we ask:

💡 *Do our findings about dataset alignment in terms of CSI hold across different budget levels?*

Figure 4 shows downstream MAE across pretraining budgets of 0.5M, 1M, 2M, and 3M samples (each trained for 5 epochs). We observe that increasing the budget for low-CSI datasets (i.e., ANI-1x and Transition-1x) beyond 2M leads to diminishing returns on rMD17 and SPICE. In contrast, increasing the pretraining budget for high-CSI datasets (i.e., OC20 and OC22) often degrades downstream performance more substantially, particularly on rMD17, SPICE, and QM9. These results indicate that allocating more compute to misaligned upstream tasks can harm downstream generalization.

**Takeaway.** Our findings are consistent across budget levels: the upstream dataset with the lowest CSI yields the best downstream performance.

## 4.3 What is the Effect of Changing the Backbone Size?

In the previous sections, we used the small variant, GemNet-OC-S, as our backbone. Here, we address the question:

💡 *Does the correlation between CSI and downstream performance hold across different backbone sizes?*

Table 2 reports the downstream performance using the large variant, GemNet-OC-L, as the backbone. We also include our best attempt at reproducing the baseline results using JMP-L pretraining (denoted as "JMP-L*").

Consistent with the results on the small backbone, models pretrained on ANI-1x achieve the best performance across all downstream tasks, aligning with its low CSI values. Notably, using a small computation budget of $\mathcal{C} = 10M$ (i.e., 2M samples over 5 epochs), ANI-1x outperforms JMP-L, which was pretrained with $\mathcal{C} = 240M$

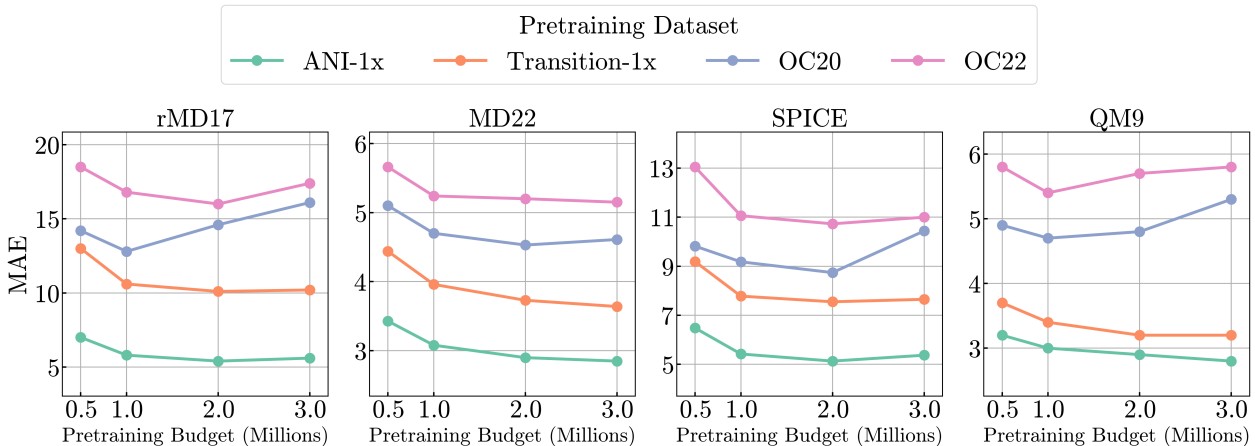

Figure 4: **Effect of Computational Budget on Performance.** While fixing the number of epochs ($\mathcal{E}$) to 5, we vary the number of training samples across $\mathcal{N} = 0.5$M, 1M, 2M, and 3M. Our findings are consistent across budget levels where the upstream dataset with the lowest CSI yields the best downstream performance.

Table 2: **Effect of Changing the Backbone Size.** We analyze the impact of using a larger variant of GemNet-OC and find that, irrespective of backbone size, relevance-based upstream dataset selection consistently outperforms costly large-scale joint pretraining.

| $\mathcal{C}$ | Upstream Data | Backbone | rMD17 (meV/Å) | MD22 (meV/Å) | SPICE (meV/Å) | QM9 (meV) |
|---|---|---|---|---|---|---|
| 240M | Joint (Temperature) | JMP-L (GemNet-OC-L) | 5.1 | 1.92 | 4.75 | 2.9 |
|  |  | JMP-L$^{*}$ (GemNet-OC-L) | 5.3 | 2.59 | 4.91 | 3.0 |
| 10M | ANI-1x | GemNet-OC-L | **4.8** | **2.54** | **5.24** | **2.6** |
|  | Transition-1x |  | 9.7 | 3.56 | 7.42 | 3.0 |
|  | OC20 |  | 13.8 | 3.90 | 9.24 | 4.6 |
|  | OC22 |  | 12.0 | 4.14 | 10.43 | 4.0 |
| 10M | Joint (Balanced) | GemNet-OC-L | 8.9 | 3.39 | 7.57 | 3.2 |
|  | Joint (Temperature) |  | 10.8 | 4.15 | 9.83 | 4.4 |

on a joint upstream dataset. We obtain state-of-the-art results with an MAE of 4.8 on Aspirin (rMD17) and 2.6 on $U_0$ (QM9), demonstrating that strong dataset alignment can outweigh large-scale pretraining even with increased model capacity. While larger backbones improve overall performance, the gap between aligned and misaligned upstream datasets persists. High-CSI datasets like OC20 and OC22 still underperform, reaffirming the importance of dataset alignment.

**Takeaway.** Our findings hold across backbone sizes: scaling up the model does not change the relative utility of upstream datasets. Alignment-based upstream dataset selection outperforms large-scale dataset mixing, even under high-capacity settings and at significantly lower computational budgets.

## 4.4 What is the Effect of Changing the Backbone Type?

In the previous experiments, we focused on the GemNet-OC backbone to enable a direct and fair comparison with the JMP baseline. We now examine whether our CSI-based alignment analysis extends beyond graph neural networks to a different architectural paradigm. Specifically, we ask:

💡 *Does the relationship between CSI and downstream performance persist across different backbone types?*

Table 3: **Effect of Changing the Backbone Type.** Downstream MAE after fine-tuning EquiformerV2 (EqV2) pretrained on different upstream datasets. Despite the architectural shift from graph neural networks to transformer-based models, upstream dataset alignment remains a strong predictor of downstream performance. Numbers in parentheses indicate parameter count.

| $\mathcal{C}$ | Pretraining Time (A100 GPU-days) | Upstream Data | Backbone | rMD17 (meV/Å) | MD22 (meV/Å) | SPICE (meV/Å) | QM9 (meV) |
|---|---|---|---|---|---|---|---|
| 10M | 1.68 | ANI-1x | EqV2 (31M) | **9.7** | **3.23** | **8.88** | 6.5 |
|  | 1.60 | Transition-1x |  | 11.0 | 3.37 | 10.16 | **6.4** |
|  | 6.78 | OC20 |  | 12.5 | 3.58 | 10.54 | 7.0 |
|  | 7.72 | OC22 |  | 12.6 | 3.72 | 11.10 | 6.7 |
| 10M | 6.69 | ANI-1x | EqV2 (87M) | **9.0** | **2.90** | **6.24** | **3.5** |
|  | 6.72 | Transition-1x |  | 10.2 | 3.05 | 7.80 | 3.6 |
|  | 16.52 | OC20 |  | 11.4 | 3.25 | 8.10 | 4.0 |
|  | 17.16 | OC22 |  | 11.9 | 3.28 | 8.60 | 4.3 |

To answer this question, we evaluate EquiformerV2 (Liao et al., 2023), a transformer-based equivariant backbone. We pretrain EquiformerV2 on the same set of upstream datasets using the identical pretraining and fine-tuning protocol as in our main experiments. We adopt the same pretraining budget of 2M samples over 5 epochs for both the small (31M parameters) and large (87M parameters) EquiformerV2 variants. Pretraining EquiformerV2 is notably more expensive than GemNet-OC. Even under this limited 10M budget, the large variant requires up to 17 A100 GPU-days. We report the results in Table 3.

Across both EquiformerV2 model sizes, performance trends remain consistent with our GemNet-OC findings. The datasets with the lowest CSI lead to the strongest downstream performance, with ANI-1x being the top-performer in nearly all cases. While Transition-1x shows a marginal advantage on QM9 for the 31M model, ANI-1x achieves the best results as capacity increases to the 87M variant. This confirms that the benefits of alignment reflect a general property of data relevance that persists across architectural paradigms.

**Takeaway.** CSI-based alignment reliably predicts downstream performance across both backbone sizes and backbone types, highlighting the central role of dataset alignment in determining pretraining performance.

### 4.5  Is More Diverse Data Always Better?

A common assumption in pretraining is that larger and more diverse datasets lead to better generalization. This intuition motivates the JMP framework, where a large-scale pretraining budget of $\mathcal{C} = 240M$ led to strong downstream results. However, it remains unclear whether this benefit comes from the size, the diversity, or the alignment of the data with the downstream task. Here, we revisit this assumption through a targeted experiment:

💡 *Does increasing data diversity by adding less aligned sources improve or harm downstream performance?*

To test this, we compare two settings: (1) pretraining on $\mathcal{N} = 2M$ unique samples from ANI-1x, the most CSI-aligned dataset, and (2) pretraining on a mixture of 2M ANI-1x samples and 1M OC22 samples (i.e., $\mathcal{N} = 3M$), both trained for 5 epochs. As shown in Figure 5, simply adding OC22 results in worse downstream performance across all four tasks, despite the increase in data volume. This indicates that adding less aligned data may interfere with the knowledge transfer gained from aligned pretraining sources.

**Takeaway.** Our results challenge the intuitive strategy of adding diversity to pretraining datasets without considering alignment. CSI provides a practical signal for curating upstream data that supports better generalization, especially under constrained budgets.

### 4.6  Is Alignment Important in Self-Supervised Pretraining?

Self-supervised learning has long been central to 3D molecular representation learning, initially driven by the scarcity of large-scale labeled data (Rong et al., 2020; Liu et al., 2021; Jiao et al., 2023; Chen et al.,

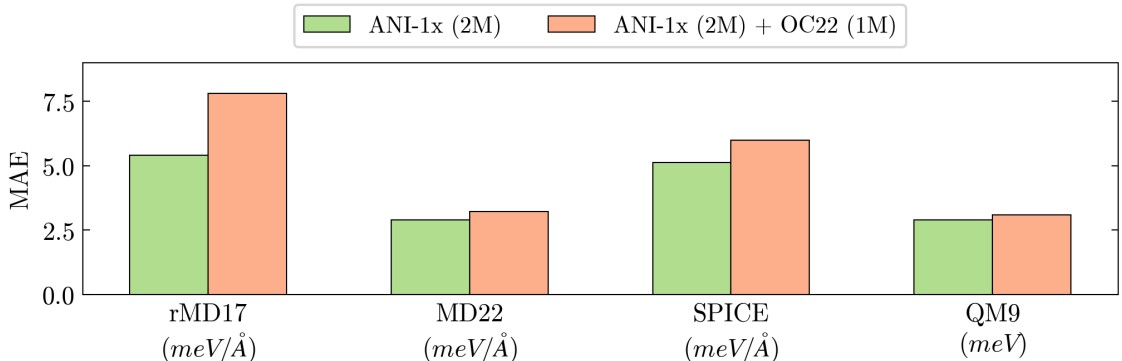

Figure 5: **Impact of Adding Less Aligned Pretraining Data.** Adding $1M$ OC22 samples to a $2M$-sample ANI-1x baseline worsens downstream performance despite a larger pretraining budget. This highlights the importance of dataset alignment and the value of the CSI metric for effective pretraining.

Table 4: **Effect of Changing the Pretraining Objective.** We report the test MAE for the EquiformerV2 (EqV2) backbone, with 87M parameters, pretrained with the DeNS self-supervised denoising objective. Consistent with our supervised results, pretraining on the most CSI-aligned dataset (i.e., ANI-1x) achieves superior fine-tuning performance across all downstream molecular tasks.

| $\mathcal{C}$ | Upstream Data | Backbone | rMD17 (meV/Å) | MD22 (meV/Å) | SPICE (meV/Å) | QM9 (meV) |
|---|---|---|---|---|---|---|
| | ANI-1x | | **4.7** | **2.78** | **5.95** | **4.1** |
| | Transition-1x | | 7.3 | 2.95 | 8.71 | 4.2 |
| 10M | OC20 | EqV2 (87M) + DeNS | 8.1 | 3.32 | 7.34 | 5.0 |
| | OC22 | | 10.4 | 3.58 | 9.39 | 4.9 |

2021; Zhou et al., 2022a; Ji et al., 2024). While recent frameworks such as JMP (Shoghi et al., 2023) emphasize supervised pretraining on energy and forces, modeling structural dependencies directly from molecular geometry remains a dominant paradigm. This raises a critical question regarding the role of data selection:

💡 *Does dataset alignment matter for self-supervised pretraining?*

To answer this, we extend our analysis to the self-supervised setting and adopt the DeNS framework (Liao et al., 2024). DeNS follows the standard denoising paradigm in 3D molecular learning, where models are trained to recover the original 3D atomic positions from noisy inputs. Unlike traditional self-supervised denoising objectives designed for equilibrium geometries (Rong et al., 2020; Zhou et al., 2022a), DeNS explicitly targets non-equilibrium molecular and material configurations. This makes it well suited to the non-equilibrium datasets considered in this work. We pretrain the EquiformerV2 (87M) backbone with the DeNS objective on each upstream dataset using a fixed budget of 10M training instances (2M samples over 5 epochs).

The results, summarized in Table 4, demonstrate that CSI remains a robust predictor of downstream performance even in the self-supervised setting. Across all tasks, the upstream dataset with the lowest CSI (i.e., ANI-1x) yields the strongest transfer performance. Remarkably, the model pretrained on ANI-1x using self-supervised denoising achieves an MAE of 4.7 meV/Å on Aspirin (rMD17), outperforming the large-scale supervised JMP-L baseline (5.1 meV/Å) while using 24× less pretraining budget. This confirms that dataset alignment remains relevant for 3D representation learning under both supervised and self-supervised objectives.

**Takeaway.** Dataset alignment is objective-agnostic. Even when the pretraining objective is changed to model structural dependencies in a self-supervised manner, CSI-aligned datasets consistently achieve superior performance.

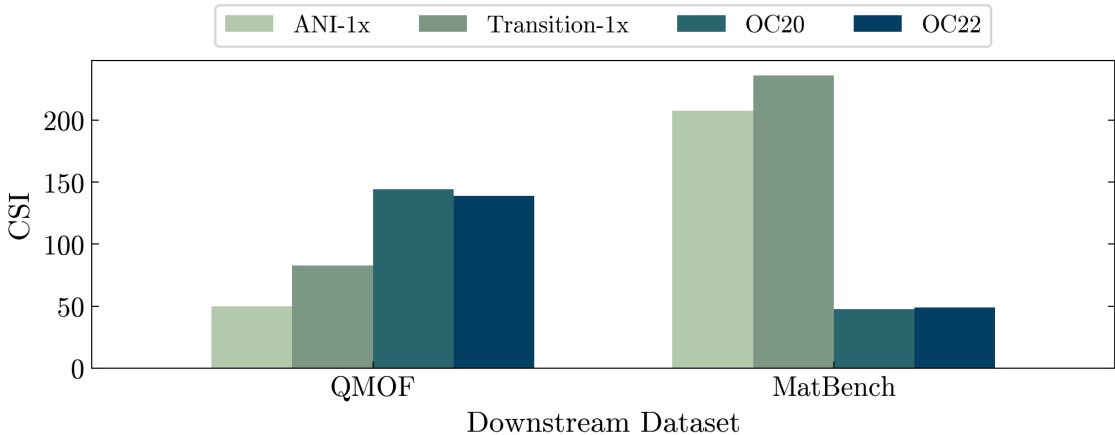

Figure 6: **CSI Between Upstream and OOD Downstream Tasks.** CSI values predict that ANI-1x is the best pretraining choice for QMOF, while OC20 and OC22 are best for MatBench.

Table 5: **OOD Task Performance Across Upstream Sources.** We compare the CSI-predicted best upstream sources with actual downstream performance on OOD tasks (QMOF, MatBench, and QM9's $\Delta_\epsilon$). While CSI aligns well with QM9's OOD label, it mispredicts the best source for MatBench. Joint pretraining generally improves performance, highlighting the benefits of diverse upstream sources for OOD generalization.

| $\mathcal{C}$ | **Upstream Data** | **Backbone** | **QM9** $[\Delta_\epsilon]$ ($meV$) | **QMOF** (eV) | **MatBench** [fold0 / mean] ($cm^{-1}$) |
|---|---|---|---|---|---|
| 240M | Joint (Temperature) | JMP-S (GemNet-OC-S) | 23.1 | 0.18 | 26.60 / 22.77 |
| | | JMP-S* (GemNet-OC-S) | 24.0 | 0.19 | 24.77 / 21.48 |
| 10M | ANI-1x | GemNet-OC-S | **24.5** | 0.22 | 30.09 / 29.60 |
| | Transition-1x | | 25.3 | 0.22 | 52.22 / 38.56 |
| | OC20 | | 30.8 | 0.22 | 37.52 / 30.88 |
| | OC22 | | 35.6 | 0.22 | 32.78 / 27.55 |
| 10M | Joint (Balanced) | GemNet-OC-S | 27.3 | **0.21** | **26.11 / 24.87** |
| | Joint (Temperature) | | 27.9 | **0.21** | 26.63 / 25.61 |

## 5 Beyond In-Distribution

Recall that our pretraining process is conducted on upstream tasks involving molecules and catalysts, with energy and force as targets. We classify as in-distribution (ID) any downstream task that uses the same labels (energy and/or forces), and as out-of-distribution (OOD) those that involve different target properties, such as band gap (e.g., QMOF) or phonon properties (e.g., MatBench). While our main results focused on ID evaluation, here we explore our metric's applicability to OOD tasks. Specifically, we examine three cases: the Band Gap property from QMOF (Rosen et al., 2021), Phonons from MatBench (Dunn et al., 2020), and $\Delta_\epsilon$ from QM9.

In Figure 6, we present the CSI values for OOD domains; for QM9, the OOD label ($\Delta_\epsilon$) uses the same dataset-level CSI values as in Figure 3. We observe that QMOF exhibits a pattern similar to other ID domains shown in Figure 3. However, MatBench displays a distinct pattern, showing strong correlation with OC20 and OC22, followed by ANI-1x and Transition-1x. Next, we analyze the correlation between CSI and downstream performance under OOD evaluation.

Table 5 shows that $\Delta_\epsilon$ in QM9 aligns with the CSI pattern, similar to ID evaluation, suggesting that CSI remains effective when the downstream domain (i.e., molecules, as in QM9) is similar to the upstream tasks, even if the label type is OOD. In QMOF, the different upstream sources achieve similar performance, which

lags behind the full pretraining by JMP. For MatBench (evaluated over 5 folds), OC22 achieves the best mean performance while OC20 lags behind, despite our metric predicting both to be equally suitable. Additionally, for both QMOF and MatBench, joint pretraining variants generalize better than individual sources. This suggests that when both the downstream domain and the label type differ from all upstream sources, mixing diverse upstream domains provides the best performance.

While CSI reliably guides dataset selection for in-distribution tasks, its effectiveness in OOD scenarios reveals important limitations of dataset-level alignment. This may stem from the limited diversity of the backbone used for feature extraction, which was pretrained only on energy and force targets. Future work could explore using backbones pretrained on broader sets of chemical properties or incorporating more diverse upstream domains to better capture variation across OOD tasks. Another promising direction is to leverage foundation models trained on multi-modal or multi-objective tasks, which may offer more transferable representations for similarity assessment across varied downstream domains.

## 6 Conclusion

This paper challenges the prevailing trend of scaling data and computational resources in atomic property prediction by demonstrating that strategic data selection based on dataset alignment can achieve comparable or superior performance with significantly fewer resources. We introduce the Chemical Similarity Index (CSI), a simple metric that quantifies the alignment between upstream pretraining datasets and downstream tasks, enabling the selection of high-quality, task-aligned pretraining data. Our experiments reveal that smaller, focused datasets often outperform larger, mixed ones, and that indiscriminately adding data can degrade performance when relevance is low. These findings highlight that alignment, rather than scale alone, is the key to effective pretraining, and they point toward a more principled, efficient, and sustainable direction for future research in atomic property prediction.

### Acknowledgment

The authors thank Mohamed Elhoseiny, Firas Laakom, and Abdelrahman Eldesokey for their valuable feedback. Yasir Ghunaim was supported by Saudi Aramco. This work is supported by the King Abdullah University of Science and Technology (KAUST) Center of Excellence for Generative AI under award number 5940 and SDAIA-KAUST Center of Excellence in Data Science, Artificial Intelligence (SDAIA-KAUST AI). The computational resources are provided by IBEX, which is managed by the Supercomputing Core Laboratory at KAUST.

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

# A   More Epochs or More Data?

To extend the findings presented in the main paper, we explore the trade-off between increasing the number of training epochs and expanding the dataset size under a fixed computational budget. Specifically, we aim to answer the following question:

💡 ***Given a fixed computational budget, is it more effective to train on a smaller dataset for more epochs or to train on a larger dataset for fewer epochs?***

**Setup.** To investigate this question, we compare two scenarios under the same computational budget of 10M samples: (1) training on 2M samples for 5 epochs, and (2) training on 1M samples for 10 epochs. We evaluate the performance of models pretrained on ANI-1x, Transition-1x, OC20, and OC22, and fine-tune them on the downstream datasets: rMD17, MD22, SPICE, and QM9. For comparison, we also include the results of JMP-L and JMP-S, which use 120M samples for 2 epochs.

**Results.** Table 6 presents the downstream performance for the two scenarios. Across all datasets, ANI-1x consistently achieves the best performance, regardless of whether the model is trained on 2M samples for 5 epochs or 1M samples for 10 epochs. For example, on rMD17, ANI-1x achieves a test error of 5.4 in both scenarios, outperforming JMP-S (6.7) and JMP-L (5.1). Similarly, on SPICE, ANI-1x achieves a test error of 5.08 (2M samples, 5 epochs) and 5.04 (1M samples, 10 epochs), compared to 5.71 for JMP-S and 4.75 for JMP-L.

Interestingly, increasing the number of epochs from 5 to 10 while reducing the dataset size from 2M to 1M does not significantly degrade performance for ANI-1x. This suggests that for highly aligned datasets like ANI-1x, training on fewer samples for more epochs can be as effective as training on more samples for fewer epochs. In contrast, for less aligned datasets such as OC20 and OC22, increasing the number of epochs only partially compensates for the reduced dataset size, as some tasks show similar performance while others experience noticeable degradation.

**Takeaway.** Our findings indicate that the trade-off between more epochs and more data depends on the alignment of the pretraining dataset with the downstream task. For highly aligned datasets like ANI-1x, training on fewer samples for more epochs can yield comparable performance. In contrast, for less aligned datasets, increasing the dataset size tends to be more beneficial. These results further show the importance of dataset quality and alignment, as quantified by CSI, in determining an effective pretraining strategy.

Table 6: Trade-off between increasing the number of samples and the number of epochs. We report the MAE for various downstream tasks while varying the pretraining sample count and epoch count simultaneously. $\mathcal{C}$, $\mathcal{N}$, and $\mathcal{E}$ denote the computational budget, number of samples, and number of epochs, respectively.

| $\mathcal{C}$ | $\mathcal{N}$ | $\mathcal{E}$ | Upstream Data | Backbone | rMD17 (meV/Å) | MD22 (meV/Å) | SPICE (meV/Å) | QM9 (meV) |
|---|---|---|---|---|---|---|---|---|
| 10M | 2M | 5 | ANI-1x | GemNet-OC-S | **5.4** | **2.90** | **5.13** | **2.9** |
| | | | Transition-1x | | 10.1 | 3.73 | 7.55 | 3.2 |
| | | | OC20 | | 14.6 | 4.53 | 8.74 | 4.8 |
| | | | OC22 | | 16.0 | 5.20 | 10.73 | 5.7 |
| 10M | 1M | 10 | ANI1x | GemNet-OC-S | **5.4** | **2.88** | **5.04** | **2.9** |
| | | | Transition1x | | 10.6 | 3.79 | 7.50 | 3.1 |
| | | | OC20 | | 14.8 | 4.67 | 10.16 | 4.9 |
| | | | OC22 | | 17.3 | 5.24 | 11.06 | 5.4 |

# B  Impact of Changing the Feature Extractor for CSI

In the main paper, we used an EquiformerV2 model pretrained on the OC20 dataset as our primary feature extractor for computing the CSI values. In this section, we explore alternative open-source pretrained EquiformerV2 checkpoints, including: ODAC23 (Sriram et al., 2024), MPtrj (Barroso-Luque et al., 2024) and MPtrj (DeNS) (Barroso-Luque et al., 2024). DeNS (Liao et al., 2024) refers to a denoising auxiliary task that extends traditional denoising to non-equilibrium structures to improve learned representations.

We report the results in Figure 7. Our comparison is based on relative rankings since the absolute values are influenced by the choice of feature extractor. Notably, for 4 out of 5 datasets (rMD17, MD22, SPICE, and OMat24), all feature extractors agree on the lowest-CSI upstream dataset: ANI-1x for rMD17, MD22, and SPICE, and OC20 for OMat24. For QM9, OC20 and MPtrj (DeNS) rank ANI-1x as the most aligned, whereas ODAC23 and MPtrj place ANI-1x second to Transition1x, which is also a strong upstream candidate for QM9.

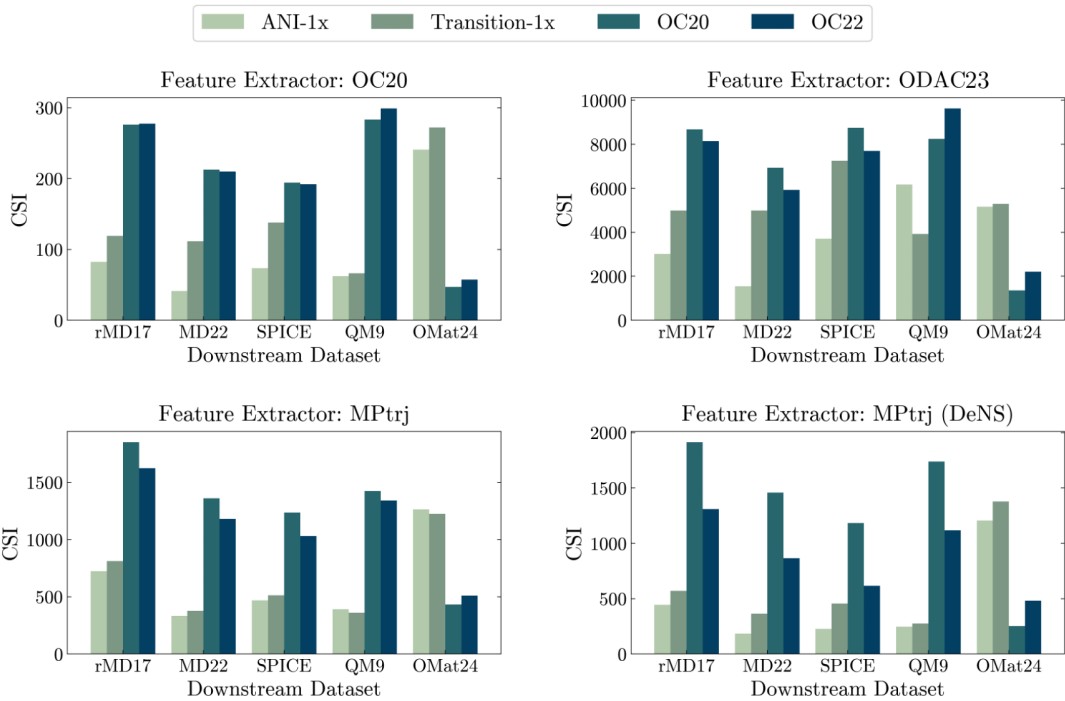

Figure 7: **Dataset Alignment Using Feature Extractors Pretrained on Various Datasets.** We compare CSI relative values computed using feature extractors pretrained on four different datasets: OC20, ODAC23, MPtrj, and MPtrj (DeNS). The lowest CSI upstream dataset is consistent across feature extractors for 4 out of 5 downstream tasks, namely rMD17, MD22, SPICE, and OMat24.

## C  Feature Representation Choices for CSI

A key challenge when adapting the Fréchet Inception Distance (FID) to 3D molecular graphs is handling their variable number of atoms. Our feature extractor, EquiformerV2, produces variable-sized node-level features in $\mathbb{R}^{n \times d}$, where $n$ is the number of atoms and $d$ is the feature dimension. A straightforward approach is to aggregate these features using mean pooling, producing a graph-level representation in $\mathbb{R}^{N \times d}$, where $N$ is the number of graphs. This setup is directly analogous to the image-based FID. The central question, however, is whether this aggregation loses valuable information.

For our CSI metric to accurately measure alignment across various domains, it is critical that the *feature expressivity* is maximized. To quantify this expressivity, we use the Effective Rank (Roy & Vetterli, 2007) of the covariance matrix of these features. The Effective Rank is similar to the traditional matrix rank but is more robust to noise.

Formally, as in Roy & Vetterli (2007), let $C \in \mathbb{R}^{d \times d}$ be the covariance matrix with eigenvalues $\{\lambda_j\}_{j=1}^d$ and $p_j = \lambda_j / \sum_{m=1}^d \lambda_m$. The Shannon (1948) entropy is $H = -\sum_{j=1}^d p_j \log p_j$, and the Effective Rank is:

$$\mathrm{erank}(C) = \exp(H).$$

Higher erank indicates that information is distributed across more independent directions, signifying a richer and less redundant feature representation.

To compare node-level and graph-level expressivity in a paired manner, we employ a bootstrap protocol. For a chosen number of graphs $k$, we proceed as follows:

1. Sample $k$ graphs from the dataset.
2. Construct two matrices in $\mathbb{R}^{k \times d}$ from the sampled graphs: (i) a graph-level matrix, where each row is the pooled representation of one graph, and (ii) a node-level matrix, where each row corresponds to a randomly selected node representation from each graph.
3. Standardize each feature (z-score per dimension) in both matrices.
4. Compute the covariance $C$ and its effective rank $\mathrm{erank}(C)$ for both variants.

We repeat this process 10 times and report the mean effective ranks. We perform this comparison for $k \in \{5{,}000, 10{,}000, 15{,}000\}$ and show the results in Figure 8. We observe that across all tested sample sizes, the node-level features consistently have a significantly larger effective rank. This finding indicates that the aggregation process for graph-level features results in a loss of intrinsic dimensions that are likely significant for dataset alignment studies. Therefore, to preserve maximum feature expressivity, we use node-level features in our CSI design.

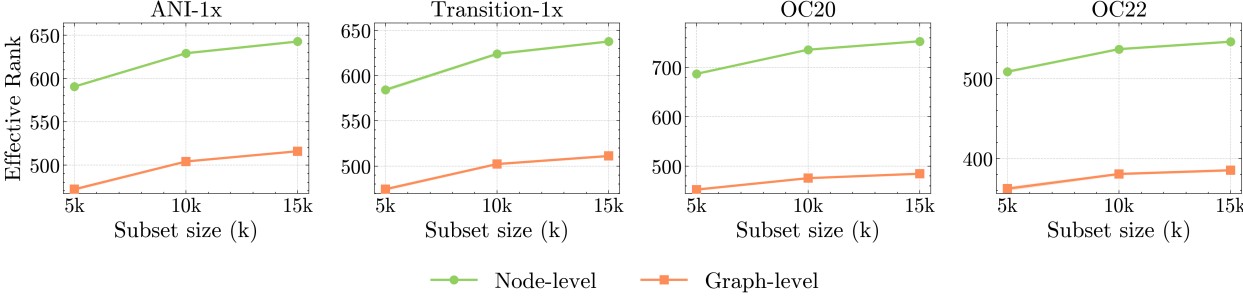

Figure 8: **Node-Level vs Graph-Level Effective Rank.** Effective rank of features from (ANI-1x, Transition-1x, OC20, OC22) datasets using both node-level (green) and mean-pooled graph-level (orange) representations. The consistently higher effective rank of node-level features demonstrates that mean pooling removes informative directions, which may be inadequate for analyzing alignment between datasets.

To complement our effective rank analysis, we compare node-level and graph-level CSI variants in Figure 9. OC20 and OC22 are both catalysis-focused datasets and share similar chemical domains. The node-level features reflect this similarity by yielding comparable scores between the two datasets. In contrast, the graph-level variant suggests greater divergence between the two, which may be due to information loss introduced by mean pooling.

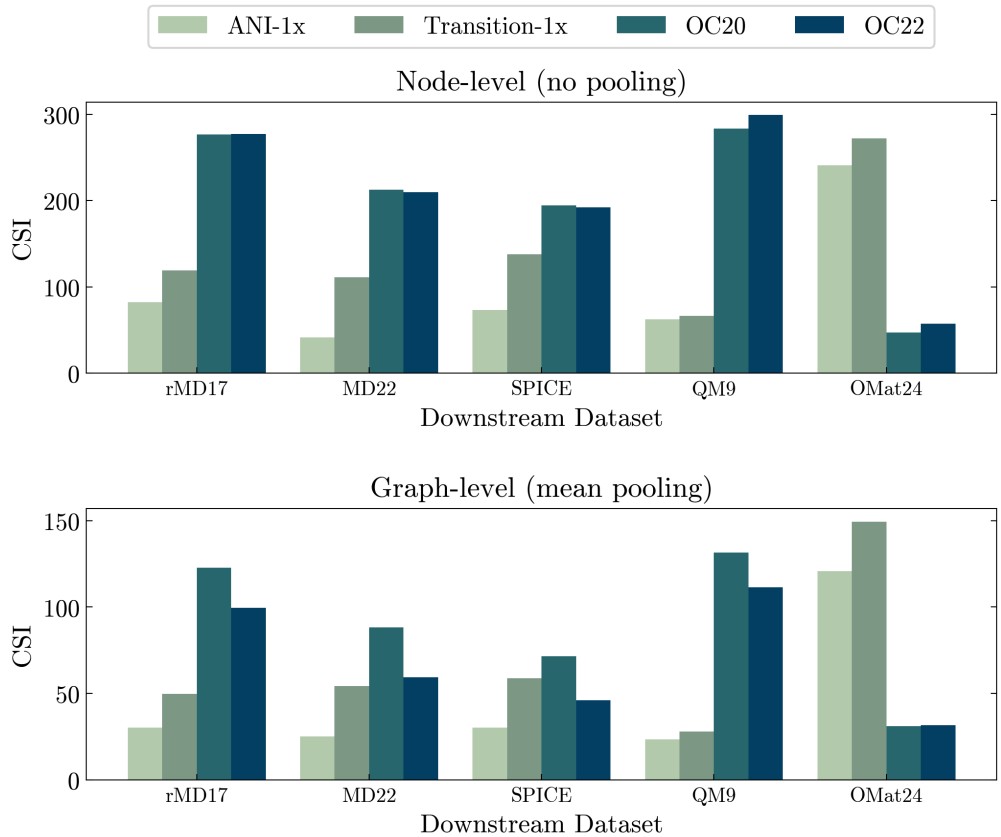

Figure 9: **CSI values for Node-Level vs Graph-Level Feature Representation**

# D Class Balancing for Feature Extraction

The upstream datasets analyzed in our main paper are large in size. For example, OC20 contains over 133 million training samples, while Transition-1x contains around 9.6 million. To make the computation of our CSI metric tractable, we sample 10K instances from each upstream dataset for feature extraction. To ensure these subsets are representative of the full distribution, we perform class-balanced sampling. For ANI-1x and Transition-1x, a class is defined by the molecular formula of the molecule in the trajectory. For OC20 and OC22, a class is defined by the bulk structure. Each class is treated as a bin, and we sample an equal number of instances from each bin.

Figure 10 compares random sampling and class-balanced sampling in terms of class coverage. Random sampling leads to significant underrepresentation of less frequent classes, while class-balanced sampling ensures uniform coverage across all classes.

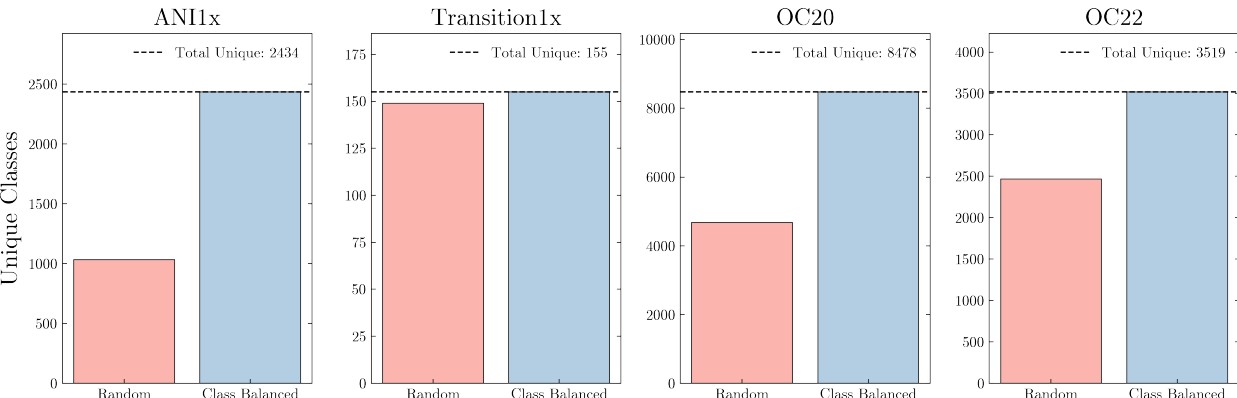

Figure 10: **Impact of Sampling Strategies on Subset Construction for Feature Extraction.** We show the differences in class coverage between random and class-balanced sampling when sampling 10K instances for each upstream task.

# E    Implementation Details

For both pretraining and fine-tuning experiments, we primarily follow the JMP hyperparameters. However, due to resource constraints requiring smaller batch sizes compared to JMP, we adjusted the learning rate to ensure training stability, as detailed below.

**For pretraining,** we use a batch size of 20 and a learning rate (LR) of 1e-4 for the small backbone (GemNet-OC-S). For the large backbone (GemNet-OC-L), the batch size is reduced to 12 to fit GPU memory. Additionally, when training with the OC22 dataset on the large backbone, a LR of 1e-4 caused gradient instability, thus we used a LR of 1e-5 for that particular run. Unless otherwise specified, each experiment is run for five epochs on the specified number of samples for each section of the paper. The best checkpoint is selected based on the performance in the validation set. To handle the large size of the upstream validation sets, validation is performed on a smaller subset of 2,000 samples.

**For finetuning,** we use the batch size specified in the JMP codebase and a default learning rate (LR) of 8e-5, except for cases where adjustments were needed to stabilize training. Specifically, we use 5e-5 for QMOF, 8e-4 for MatBench when pretrained on Transition1x.

