# OpenReview forum: "On the Importance of Pretraining Data Alignment for Atomic Property Prediction"
_TMLR — Accepted by TMLR_

### Review · Reviewer_yRFh · 2025-07-15

**Summary Of Contributions:**

For atomic‐scale property prediction, the work claims that carefully choosing a pre‑training dataset that is chemically aligned with the downstream task can outperform blindly scaling to hundreds of millions of structures. Using 2M ANI‑1x molecules, the authors match or beat SOTA MAE on 4 standard benchmarks with lower pre-training computation resources with certain pre-training schema and backbone model structures. Ablation studies show that lower distance from training and test datasets correlates with better in‑distribution accuracy at any compute budget.

**Audience:**

Yes

**Claims And Evidence:**

No

**Requested Changes:**

- Using a proper pre-training objective (e.g., those used in the referred papers).
- Using Transformers as the backbone model.
- Adding experiments and analysis on open-source pre-trained molecular representation models.

**Strengths And Weaknesses:**

## Strengths

The paper shows that small but task‑aligned data beats scaled pre-training dataset, using just 2 M ANI‑1x molecules to rival a 120 M‑sample baseline while reducing compute. It introduces the simple, model‑agnostic metric Chemical Similarity Index (CSI) that predicts transfer quality, and validates its claims across budgets, model sizes and both in‑ and out‑of‑distribution tasks. The paper reveals how misaligned extra data can hurt.

## Weaknesses

In my opinion, **the pre-training objective is wrong**. I know that there are previous works using the same pre-training scheme, but it does not necessarily validate it's theoretical correctness. From my perspective, the pre-training objective should focus on modeling the structural dependencies within the molecules, rather than performing any property prediction. The authors could check out the pre-training methods adopted by Grover or Uni-Mol [1,2]. This is a fundamental flaw of this work and it makes the soundness of all subsequent discussion questionable.

Additionally, the backbone model architecture is somewhat outdated. Most of the SOTA works are using Transformers, which have been proven to be more effective in capturing the data features. Experimenting on such backbone models might deliver different results.

One other concern is related to the usage of the pre-trained models. In both NLP and materials science domains, people tend to directly apply fine-tuning on open-source pre-trained models, rather than manually go through the pre-training process themselves. With this pipeline, the demand on computation resources is not as significant as what have been proposed in the paper.


[1] Zhou, Gengmo, et al. "Uni-mol: A universal 3d molecular representation learning framework." (2023).
[2] Rong, Yu, et al. "Self-supervised graph transformer on large-scale molecular data." Advances in neural information processing systems 33 (2020): 12559-12571.

---

> ### Author Response · Authors · 2025-08-14
>
> We thank the reviewer for their feedback. We address each point in detail below.
>
> >In my opinion,  **the pre-training objective is wrong**. I know that there are previous works using the same pre-training scheme, but it does not necessarily validate it's theoretical correctness. From my perspective, the pre-training objective should focus on modeling the structural dependencies within the molecules, rather than performing any property prediction. The authors could check out the pre-training methods adopted by Grover or Uni-Mol [1,2]. This is a fundamental flaw of this work and it makes the soundness of all subsequent discussion questionable.
>
> We appreciate the reviewer’s perspective. While we acknowledge that alternative self-supervised objectives such as those in GROVER and Uni-Mol exist, we respectfully disagree that using supervised pretraining is a fundamental flaw.
>
> First, our work does not propose a new pretraining objective; instead, we focus on improving **pretraining dataset selection** within established 3D frameworks. We adopt the JMP (ICLR 2024) setup because (i) supervised energy/force pretraining is competitive on 3D molecular/material tasks, (ii) it supports non-equilibrium structures common in large-scale molecular/material datasets, and (iii) it includes diverse upstream sources, making it a natural baseline for domain-alignment analyses.
>
> Second, from a theoretical perspective, energies and forces are one of the most **fundamental** physical quantities of molecular systems as governed by the Schrödinger equation (non-relativistic, time-independent scenario) [3], and many other molecular properties can be derived from them. Empirical evidence further supports this: the JMP paper includes a comparison with the self-supervised Pre-training via Denoising [6] (which uses a denoising strategy similar to Uni-Mol [1]). As shown in Table 2 of the JMP paper [4], supervised pretraining outperforms the self-supervised Pre-training via Denoising (GNS+TAT+NN) [6], which motivated our focus on JMP.
>
> Third, many self-supervised objectives are either 2D (e.g., GROVER [2] on SMILES) or primarily evaluated on equilibrium 3D geometries (e.g., Uni-Mol [1], Uni-Mol2 [5], Pretraining via Denoising [6] ). Our upstream datasets (OC20, OC22, ANI-1x, Transition1x) contain mostly non-equilibrium structures, where traditional denoising has recently been shown to be ill-posed for such configurations [7]. Recent work DeNS [7] addresses this but still requires force labels, making it partially supervised. Given these constraints, supervised pretraining is currently the most practical and effective choice for large-scale 3D tasks.
>
> We have updated the Related Work section to include these clarifications in the revised manuscript.
>
> - [1] Zhou, Gengmo, et al. "Uni-mol: A universal 3d molecular representation learning framework." (2023).
> - [2] Rong, Yu, et al. "Self-supervised graph transformer on large-scale molecular data." Advances in neural information processing systems 33 (2020): 12559-12571.
> - [3] Sholl, David S., and Janice A. Steckel. _Density functional theory: a practical introduction_. John Wiley & Sons, 2022.
> - [4] Shoghi, Nima, et al. "From Molecules to Materials: Pre-training Large Generalizable Models for Atomic Property Prediction." _The Twelfth International Conference on Learning Representations_.
> - [5] Ji, Xiaohong, et al. "Uni-mol2: Exploring molecular pretraining model at scale." NeurIPS (2024).
> - [6] Zaidi, Sheheryar, et al. "Pre-training via denoising for molecular property prediction." ICLR (2023).
> - [7] Liao, Yi-Lun, et al. "Generalizing denoising to non-equilibrium structures improves equivariant force fields." TMLR (2024).

---

> ### Author Response · Authors · 2025-08-14
>
> >Additionally, the backbone model architecture is somewhat outdated. Most of the SOTA works are using Transformers, which have been proven to be more effective in capturing the data features. Experimenting on such backbone models might deliver different results.
>
> We primarily used GemNet-OC to match the backbone used in JMP for direct comparison. To address the reviewer’s concern, we now include additional experiments using the transformer-based EquiformerV2 (ICLR 2024) backbone in Appendix B. We note that the results are consistent with our main findings confirming that our main hypothesis holds for different backbones.
>
> >One other concern is related to the usage of the pre-trained models. In both NLP and materials science domains, people tend to directly apply fine-tuning on open-source pre-trained models, rather than manually go through the pre-training process themselves. With this pipeline, the demand on computation resources is not as significant as what have been proposed in the paper.
>
> The reviewer raises an important point. Indeed, most current research uses publicly available pretrained models. However, the increasing computational cost of pretraining has made the development of new pretrained models accessible only to those with substantial resources. Our findings demonstrate that it is still possible to contribute meaningfully to pretraining research with limited resources. For example, while JMP required 34,400 and 57,000 V100 GPU-hours to pretrain JMP-L and JMP-S respectively, our pretraining on ANI-1x (2M) used only 24 V100 GPU-hours for the small backbone and 55 V100 GPU-hours for the large backbone. This corresponds to a reduction in compute cost by factors of 1433× and 1036×, respectively, which makes pretraining development significantly more affordable and accessible.
>
> We have clarified this point in the Introduction of the revised manuscript to better highlight the computational costs of pretraining in 3D tasks.

---

> > ### Comment · Reviewer_yRFh · 2025-08-16
> >
> > Thank you for your comment.
> >
> > I realize I may not have articulated my first point clearly. I do agree that supervised pre-training can have merit in certain cases. However, the challenge with such an objective is that the knowledge acquired is often not easily transferable across tasks. When the downstream task distribution only marginally overlaps with the pre-training data, the benefit from pre-training is minimal, leading naturally to the conclusion you reached. In contrast, pre-training on molecular structure prediction inherently introduces a broader distribution of information, which is more likely to benefit a wider range of downstream tasks. For this reason, I believe a more substantive set of experiments should focus on the latter rather than the former setup.
> >
> > I also appreciate the additional experiments you conducted in response to my second point. That said, I find the argument for the third point less convincing. As noted above, a well-designed pre-training objective should equip the model with general molecular knowledge that consistently contributes to downstream performance. In fact, in practical scenarios, one can freeze the backbone foundation model and fine-tune only the final prediction layer. This approach is not only effective but also extremely cost-efficient. A large amount of experiments should also be conducted to address this concern.
> >
> > Considering the above, I believe the paper still requires a major revision.

---

> > > ### Comment · Reviewer_yRFh · 2025-08-16
> > >
> > > Including reference [1] to support my argument. The authors can refer to Figure 5 for a comparison of the different behaviors of models pre-trained with various objectives.
> > >
> > > [1] Li, Yinghao, et al. "Muben: Benchmarking the uncertainty of molecular representation models." arXiv preprint arXiv:2306.10060 (2023).

---

> > > ### Author Response · Authors · 2025-08-19
> > >
> > > >I realize I may not have articulated my first point clearly. I do agree that supervised pre-training can have merit in certain cases. However, the challenge with such an objective is that the knowledge acquired is often not easily transferable across tasks. When the downstream task distribution only marginally overlaps with the pre-training data, the benefit from pre-training is minimal, leading naturally to the conclusion you reached. In contrast, pre-training on molecular structure prediction inherently introduces a broader distribution of information, which is more likely to benefit a wider range of downstream tasks. For this reason, I believe a more substantive set of experiments should focus on the latter rather than the former setup.
> > >
> > > We appreciate your clarification. However, we want to reiterate that our use of supervised pretraining follows a well-established framework, JMP [4], which was peer-reviewed, accepted at ICLR 2023, and has been cited by 65 papers. **Our work does not introduce a new pretraining objective.** Instead, it studies how dataset alignment affects model performance under a commonly used setup.
> > >
> > > We respectfully ask you to focus on the strengths and weaknesses of **our contribution**. The choice of pretraining objective follows existing standards in the field. For clarity, we reiterate two key points:
> > >
> > >  ---
> > >
> > > **1. Supervised Pretraining Can Be Effective Under Distribution Shift**
> > >
> > > JMP [4] is trained on energy and force labels and has demonstrated strong performance on out-of-distribution datasets such as QMOF and MatBench. These benchmarks contain unseen chemical structures and target labels that differ from the pretraining objectives. This supports the idea that supervised learning grounded in physical properties can generalize even when the overlap between pretraining and downstream distributions is limited.
> > >
> > > **2. Self-Supervised Learning for Non-Equilibrium Structures Remains an Open Problem**
> > >
> > > Our work focuses on non-equilibrium structures, as in JMP [4], and as found in large-scale datasets like ANI-1x, Transition-1x, OC20, and OC22. As discussed in [7], applying self-supervised methods in this context is still an open research problem. Supervised pretraining remains a practical and widely adopted choice in this setting.
> > >
> > >
> > > > I also appreciate the additional experiments you conducted in response to my second point. That said, I find the argument for the third point less convincing. As noted above, a well-designed pre-training objective should equip the model with general molecular knowledge that consistently contributes to downstream performance. In fact, in practical scenarios, one can freeze the backbone foundation model and fine-tune only the final prediction layer. This approach is not only effective but also extremely cost-efficient. A large amount of experiments should also be conducted to address this concern.
> > >
> > > We would like to clarify that your comment assumes the existence of a pretrained model. However, our work focuses on how to obtain that pretrained model in the first place. The choice of whether to perform linear probing, full fine-tuning, or freezing the backbone is a downstream decision and lies outside the scope of our contribution.
> > >
> > > Our objective is to make pretraining more accessible and efficient by identifying better-aligned datasets. This is a prerequisite for any downstream usage, including the strategy you mentioned. Therefore, the argument does not undermine our work, which directly addresses the cost and feasibility of producing strong foundation models in the first place.
> > >
> > >
> > > **References**
> > >
> > > [4] Shoghi et al., “From Molecules to Materials: Pre-training Large Generalizable Models for Atomic Property Prediction,” ICLR 2023.
> > >
> > > [7] Liao et al., “Generalizing denoising to non-equilibrium structures improves equivariant force fields,” TMLR 2024.

---

### Review · Reviewer_rtE1 · 2025-07-20

**Summary Of Contributions:**

This work aims to determine whether, in atomic property prediction, we can achieve equal or better results by pretraining on much smaller datasets that are more closely aligned with the target chemical space. The results indicate that pretraining on high-quality and task-aligned datasets can produce similar outcomes when training on only a fraction of the data used currently to train large atomic models. The authors introduce a chemical similarity index (CSI) inspired by the FID score from computer vision. The GemNet-OC-S model is pretrained on four molecular datasets, where its task is to predict energy and per-atom forces, and then fine-tuned on four downstream datasets. The CSI between the upstream and downstream datasets correlates with the model's performance. Additionally, the authors explore how changing the backbone model size and training on three less related tasks affect the results.

**Audience:**

Yes

**Broader Impact Concerns:**

The paper does not mention any broader impact concerns. Since it is a study on pretraining machine learning models to enhance performance on atomic property prediction tasks, a detailed discussion of ethical issues is not required.

**Claims And Evidence:**

Yes

**Requested Changes:**

Please see the weaknesses above. The most important changes that would improve the impact of the paper are:
1. Publishing the code to enhance the reproducibility of this research.
2. Finding more examples of downstream datasets where ANI-1x is not the most similar dataset. Currently, some upstream datasets consistently perform well, while others perform poorly.
3. Repeating the experiments on multiple folds and providing confidence intervals would strengthen the results.
4. Considering also using other models to compute CSI since the current model is pretrained on one of the upstream tasks.

**Strengths And Weaknesses:**

Strengths:
1. The paper is very clear and well-organized, presenting specific scientific questions and supporting them with proper experimental evidence. The background information and definitions provide a foundation for understanding the paper.
2. The study addresses an important issue: how to conduct effective and inexpensive pretraining for molecular models to achieve optimal results.
3. The experiments are well-designed and investigate how different variables, such as computational budget, model size, and task relevance, impact performance on downstream tasks.

Weaknesses:
1. The code is not provided, which limits the applicability of the presented method. In particular, the implementation of the CSI would help practitioners find the most relevant pretraining datasets for their projects. As the paper may have broader implications, the code would increase the reproducibility of this study and enable the confirmation of these results on other datasets.
2. The features extracted for calculating the CSI are based on a model trained on OC20, one of the evaluated pre-training datasets. Using feature extractors trained on different datasets would be necessary to make sure the results are not biased towards this dataset.
3. Based on Figure 3, all downstream datasets share a similar CSI ranking with respect to pre-training datasets. For all datasets, ANI-1x is the most similar dataset, and the two OC datasets are the least similar. It is unclear whether the good performance of the models pretrained on ANI-1x stems from high similarity or other qualities of this particular dataset.
4. No confidence intervals are provided for the results in Table 1, which makes it difficult to estimate the significance of these results.
5. Why are the results of pre-training on the combined datasets, with a small budget, not provided in Table 2? In this case, the results of the models pre-trained on the joint dataset with a 240M budget outperform those of models pre-trained on a single dataset (first row, MD22 and SPICE).
6. The results in Table 3 again suggest that ANI-1x is a generally good pre-training dataset, yielding good performance even when the downstream dataset is not similar to it (MatBench). This contradicts the main claim of the paper.

---

> ### Author Response · Authors · 2025-08-14
>
> We thank the reviewer for the insightful feedback. Below, we address each comment in turn.
>
> >Publishing the code to enhance the reproducibility of this research.
>
> We are preparing the code for release and will make it public upon acceptance.
>
> >Finding more examples of downstream datasets where ANI-1x is not the most similar dataset. Currently, some upstream datasets consistently perform well, while others perform poorly.
>
> In response to this request, we added the CSI scores and fine-tuning results for the OMat24 [1] dataset, a recently released materials dataset from Meta FAIR. We updated Figure 3 to include the OMat24 CSI score and Table 1 to report the corresponding fine-tuning results. Figure 3 shows that OC20 and OC22 are the most aligned datasets for OMat24. Fine-tuning on OMat24 with models pretrained on OC20 or OC22 (Table 1) yields better performance than with models pretrained on ANI-1x or Transition-1x. This further confirms the robustness of our proposed approach.
>
> [1] Barroso-Luque, Luis, et al. "Open materials 2024 (omat24) inorganic materials dataset and models." arXiv preprint arXiv:2410.12771 (2024).
>
> >Repeating the experiments on multiple folds and providing confidence intervals would strengthen the results.
>
> We have updated Table 1 to report the mean and standard deviation over 3 runs with different random seeds. The updated results are consistent with our initial findings on the alignment between upstream and downstream datasets.
>
> >Considering also using other models to compute CSI since the current model is pretrained on one of the upstream tasks.
>
> We have revised the manuscript to include new results for CSI using three additional pretrained feature extractors. The analysis and results are provided in Appendix C and discussed in the main body of the revised manuscript. The results demonstrate consistent relative rankings across most downstream tasks.
>
> >Why are the results of pre-training on the combined datasets, with a small budget, not provided in Table 2? In this case, the results of the models pre-trained on the joint dataset with a 240M budget outperform those of models pre-trained on a single dataset (first row, MD22 and SPICE).
>
> Since Table 1 already demonstrates that selective pretraining outperforms joint pretraining under the same compute budget, we did not initially include joint pretraining results in Table 2 given the higher computational cost of training with the larger backbone. To address this, we have updated Table 2 to include results for Joint Balanced-based sampling. The Joint Temperature-based experiments are still in progress and will be added once completed. Similar to Table 1, we expect Temperature-based to underperform Balanced-based, as it is more skewed toward less aligned upstream datasets (i.e., OC20/OC22). In general, among the limited-budget baselines (10M), the results still show that individual pretraining is superior to joint pretraining.
>
> Regarding MD22 and SPICE performance, we used the reproduced JMP-L* as our baseline for a fair comparison. We obtained JMP-L* by running the original JMP code and fine-tuning the official JMP open-source checkpoints. The ANI-1x pretrained model achieved 2.54 on MD22 compared to 2.59 for JMP-L* while performing slightly worse on SPICE.
>
> >The results in Table 3 again suggest that ANI-1x is a generally good pre-training dataset, yielding good performance even when the downstream dataset is not similar to it (MatBench). This contradicts the main claim of the paper.
>
> The objective of Section 5 is to discuss the limitations of our proposed alignment-based dataset selection. In this case, MatBench phonon prediction is classified as out-of-distribution (OOD) since both the chemical domain (bulk materials) and the label type (phonon properties) differ from the pretraining datasets. We hypothesize that in such settings, the energy/force-pretrained feature extractor may not accurately capture the feature distribution relevant for the non-energy/force property. We believe this limitation could be addressed as more universal feature extractors become available to the community.
>
> Regarding the comment that “ANI-1x is a generally good pre-training dataset,” we added new results in Table 1 for OMat24, which contains bulk materials similar to MatBench but with energy and force labels. Since both the upstream datasets and OMat24 share the same property type (energy/forces), we treat OMat24 as an in-distribution evaluation. The results show that pretraining on OC20 or OC22 outperforms pretraining on ANI-1x, consistent with our CSI predictions (Figure 3).

---

> > ### Comment · Reviewer_rtE1 · 2025-08-16
> >
> > Thank you for your responses, which address many of the initial comments. I especially appreciate the additional experiments with the OMat24 dataset, as they reinforce the paper's main claim. Additionally, I believe that experiments with different feature extractors and confidence intervals improve the study's overall robustness. I have a few follow-up comments or questions after reviewing other reviews and responses.
> >
> > 1. The new Appendix D, created in response to Reviewer 1xGw's comments, lacks an objective criterion for choosing the appropriate CSI formula. Currently, the selection seems to be based on the better correlation obtained in Table 1, which biases the similarity metric toward the expected experimental results (hypothesis). The CSI metric should be chosen objectively by defining the qualities it should have and comparing two versions based on these qualities, rather than on how well they align with the research hypothesis.
> > 2. I appreciate the new results in Tables 1 and 2. I hope the confidence intervals for the new dataset in Table 1 and the numbers for the temperature-based pretraining can be finalized before publication.
> > 3. Thank you for providing CSI scores for different checkpoints of EquiformerV2 in Appendix C. These rankings are generally consistent, but the magnitude of CSI values varies between checkpoints, especially for the ODAC25 extractor on the QM9 dataset. Do you have any ideas why, in this case, the CSI values for the ANI-1x dataset are so high? Since these values seem to be biased by the model used, do you think it would be possible to create a model-agnostic node embedding using, for example, an ensemble of randomly initialized models or handcrafted node/neighborhood features instead?

---

> > > ### Author Response · Authors · 2025-08-18
> > >
> > > We thank the reviewer for the thoughtful and engaging discussion.
> > >
> > > >The new Appendix D, created in response to Reviewer 1xGw's comments, lacks an objective criterion for choosing the appropriate CSI formula. Currently, the selection seems to be based on the better correlation obtained in Table 1, which biases the similarity metric toward the expected experimental results (hypothesis). The CSI metric should be chosen objectively by defining the qualities it should have and comparing two versions based on these qualities, rather than on how well they align with the research hypothesis.
> > >
> > > We agree that the original formulation of Appendix D did not sufficiently justify our design choice. The appendix aimed to provide additional insight into the behavior of different CSI formulations, but we acknowledge that it lacked an objective criterion for comparison.
> > >
> > > To address this, we have revised Appendix D to include a new quantitative analysis of feature expressivity at both the node and graph levels. Using effective rank [1], an entropy-based metric [2], we measure the number of independent dimensions for each representation type. A higher effective rank indicates a richer, less redundant feature representation. Node-level features consistently show higher effective rank than graph-level ones across all upstream datasets. This highlights that node-level representations are more expressive for capturing 3D molecular distributions and offer stronger signals for dataset alignment.
> > >
> > > [1] Roy, Olivier, and Martin Vetterli. "The effective rank: A measure of effective dimensionality." 2007 15th European signal processing conference. IEEE, 2007.
> > > [2] Shannon, Claude E. "A mathematical theory of communication." The Bell system technical journal 27.3 (1948): 379-423.
> > >
> > > >I appreciate the new results in Tables 1 and 2. I hope the confidence intervals for the new dataset in Table 1 and the numbers for the temperature-based pretraining can be finalized before publication.
> > >
> > > The experiments are currently running, and will reflect them in the paper as soon as possible.
> > >
> > >
> > > >These rankings are generally consistent, but the magnitude of CSI values varies between checkpoints, especially for the ODAC25 extractor on the QM9 dataset. Do you have any ideas why, in this case, the CSI values for the ANI-1x dataset are so high? Since these values seem to be biased by the model used, do you think it would be possible to create a model-agnostic node embedding using, for example, an ensemble of randomly initialized models or handcrafted node/neighborhood features instead?
> > >
> > > Since CSI is based on the Fréchet Inception Distance (FID), it is inherently influenced by the choice of feature extractor. The ODAC23 (mistyped as ODAC25 in our appendix) dataset contains metal-organic frameworks (MOFs) with a distinct topology that differs substantially from the other datasets we considered.  This distribution shift likely causes the feature extractor to exaggerate the representational differences, leading to the observed variation.
> > >
> > > As formulated in Section 3.1, we interpret CSI as a measure of relative distance rather than absolute magnitude. Given a set of candidate datasets, CSI is designed to identify the one most similar to a target dataset of interest. We agree that exploring model-agnostic embeddings is a promising direction that could further strengthen CSI by reducing model bias and complementing its current design.

---

> > > > ### Author Response · Authors · 2025-08-19
> > > >
> > > > We appreciate the constructive discussion, which has helped strengthen our work. If you have any further questions, we would be happy to address them. If you believe all your concerns have been resolved, we kindly invite you to update your evaluation of the claims and supporting evidence to "Yes."

---

> > > > > ### Author Response · Authors · 2025-08-19
> > > > >
> > > > > Thank you for your patience. As requested, we have updated Table 2 with the results of temperature-based joint pretraining using the large backbone. Consistent with Table 1, we observe that balanced sampling outperforms temperature-based sampling, as the latter skews the joint distribution toward samples from OC20 and OC22, which are not aligned with the downstream tasks.
> > > > >
> > > > > We will provide the remaining experiments on multiple seeds for OMat24 as soon as they finish running.

---

> > > ### Author Response · Authors · 2025-08-22
> > >
> > > >I appreciate the new results in Tables 1 and 2. I hope the confidence intervals for the new dataset in Table 1 and the numbers for the temperature-based pretraining can be finalized before publication.
> > >
> > > We have now included the remaining results, namely the confidence intervals for OMat24 in Table 1 and the fine-tuning results using temperature-based pretraining in Table 2.

---

### Review · Reviewer_1xGw · 2025-08-08

**Summary Of Contributions:**

This paper investigate the importance of data selection between pretraining and downstream tasks. The authors proposed a data alignment metric, called Chemical Similarity Index, to quantify dataset level similarity. According to this study, selecting small but aligned data for pretraining can achieve similar or better performance in downstream tasks than using large but less related data in in-distribution tasks.

**Audience:**

Yes

**Claims And Evidence:**

Yes

**Requested Changes:**

1. In the details of CSI implementation, the authors mention the node embeddings are used as features and class-balanced sampling is used to ensure representative coverage of molecular type.  1) Is the feature of a molecule represented as $\mathbf{X} \in \mathbb{R}^{n \times d} , n=\text{num of nodes}, d=\text{feature dimension}$? If that is the case, when generating the feature matrix of the entire dataset and calculating the mean and covariance for FID calculation, larger graphs and smaller graphs will have unbalanced contribution. 2) How is class-balanced sampling achieved?

2. In Section 4 downstream datasets part, the author says "Given the large number of pairs for ID evaluation, we focus on the first molecule for force tasks and the energy target for the multi-property dataset QM9". What does this sentence mean?

3. In Section 4 baselines part, JMP-S* and JMP-L* are reproduced finetuning results using the official JMP checkpoints. Are they finetuned using the JMP training data?

4. Can you compare CSI with other dataset similarity metric such as Fréchet ChemNet Distance?

**Strengths And Weaknesses:**

Strengths
1. this paper conduct in-depth study of the effects of alignment between pretraining and finetuning datasets on downstream task performance.

Weaknesses
1. FID-like metric has been proposed to quantify dataset similarity for molecules [1] and used in molecular benchmarks [2].
2. The conclusion of this paper, similar data for pretraining and finetuning yields better results, seems to be well known and trivial. What is the best approach to quantify molecular dataset similarity is valuable, but this paper did not compare with other similarity metrics.

[1] Preuer, Kristina, et al. "Fréchet ChemNet distance: a metric for generative models for molecules in drug discovery." Journal of chemical information and modeling 58.9 (2018): 1736-1741.
[2] Brown, Nathan, et al. "GuacaMol: benchmarking models for de novo molecular design." Journal of chemical information and modeling 59.3 (2019): 1096-1108.
[3]

---

> ### Author Response · Authors · 2025-08-14
>
> We appreciate the reviewer’s feedback. We address the points raised below:
>
>
> > In the details of CSI implementation, the authors mention the node embeddings are used as features and class-balanced sampling is used to ensure representative coverage of molecular type. Is the feature of a molecule represented as  $\mathbf{X} \in \mathbb{R}^{n \times d},\ n = \text{num of nodes},\ d = \text{feature dimension}$?
>
> Yes, that is correct.
>
> > If that is the case, when generating the feature matrix of the entire dataset and calculating the mean and covariance for FID calculation, larger graphs and smaller graphs will have unbalanced contribution.
>
> We understand the reviewer’s concern, and it is something we considered when designing our method. We chose to compute FID over node embeddings rather than pooled graph embeddings to maintain the contribution of individual atoms and avoid the potential oversmoothing that mean pooling can introduce.
>
> To test this hypothesis, we added Appendix D in the revised manuscript comparing node-level and graph-level feature representations. Using node-level features, the alignment ranking between upstream and downstream tasks (Figure 8) closely matches the performance ranking in Table 1. In contrast, graph-level features produce less consistent alignment with downstream results. These findings support our decision to preserve node-level features for CSI computation.
>
> > How is class-balanced sampling achieved?
>
> We achieve class-balanced sampling by treating each class as a bin and sampling an equal number of instances from each bin. Our definition of a class depends on the dataset. For molecules such as ANI-1x and Transition-1x, a class is defined by the molecular formula of the molecule in the trajectory. For catalysis datasets such as OC20 and OC22, a class is defined by the bulk structure. Since the upstream datasets are large (e.g., OC20 contains over 133 million training samples and Transition-1x around 9.6 million), we subsample 10K instances from each dataset for feature extraction to manage computational cost. Class-balancing ensures that our subset covers all molecule/material types. To further clarify this, we added Appendix E in the revised manuscript with more details on our class-balancing procedure.
>
> >In Section 4 downstream datasets part, the author says "Given the large number of pairs for ID evaluation, we focus on the first molecule for force tasks and the energy target for the multi-property dataset QM9". What does this sentence mean?
>
> We agree with the reviewer that the phrasing is unclear. What we meant is that, to keep the experiments tractable, we limit the evaluation to a single target from each downstream task, since we run six baselines for every dataset. For example, rMD17 contains 10 targets, so evaluating all would require 6×10=60 experiments. To avoid any selection bias, we choose the first target listed in the JMP table for force datasets (i.e., rMD17, MD22, and SPICE) and the energy property ($U_0$) for QM9. We have clarified this sentence in the revised manuscript.
>
> >In Section 4 baselines part, JMP-S* and JMP-L* are reproduced finetuning results using the official JMP checkpoints. Are they finetuned using the JMP training data?
>
> Yes, we obtained JMP-S* and JMP-L* by loading the official JMP checkpoints in the original JMP code, and then fine-tuned them on each downstream task using the same code for dataset downloading and preprocessing. All our codebase is based on JMP.
>
> > Can you compare CSI with other dataset similarity metric such as Fréchet ChemNet Distance?
>
> Both CSI and Fréchet ChemNet Distance (FCD) are derived from the Fréchet Inception Distance. However, there are key differences between the two approaches. First, FCD operates on SMILES-based molecular representations and is primarily used to evaluate the quality of generated molecules within the same domain. In contrast, CSI is designed for 3D graph-structured data where features are defined on atoms and their spatial relationships, and the data may include both equilibrium and non-equilibrium structures. Second, CSI is the first in the molecular field to adapt the FID framework to measure the alignment between upstream and downstream datasets across several domains (e.g., molecular and materials domains) while FCD evaluates molecular generative models in a single domain. Therefore, we cannot directly compare with FCD, and we are not aware of other metrics directly comparable to CSI.
>
> We have revised the manuscript to include a detailed discussion on distribution similarity in the related work section, citing [1] and [2].

---

> > ### Author Response · Authors · 2025-08-19
> >
> > Thank you for your thoughtful engagement. Your feedback has been instrumental in improving the paper. If there are any remaining questions or concerns, we would be glad to respond.

---

### Decision · Action_Editor_hgmZ · 2025-12-05

**Recommendation:** Accept with minor revision

**Audience:**

Yes

**Audience Explanation:**

Compared to the quality and diversity, the alignment of pretrained data is less studied in the community. This paper provides one data point to support this.

**Claims And Evidence:**

No

**Claims Explanation:**

After rebuttal, most reviewers appreciate the contribution of the work in general. I would however agree with some points raised by the reviewers, and urge the authors to revise the paper for the final version by adding more experiments or discussions on the following problems:
1. While the paper focuses on a specific setting of supervised learning, the scope is limited thus making the paper less interesting for audience. A more convincing way to support the conclusion is to go beyond this setting by considering the convention unsupervised pretraining setting such as those used contrastive learning. Please add discussions or preliminary results (if any) on whether this conclusion from this paper apply to other settings.
2. Results with more backbones (e.g., Transformer), data sizes (increase the data size and ablate the impact of data sizes) and model sizes (ablate with different model sizes) should be included. The current paper is limited to one backbone on a relatively small data size, making the generalization of the finding less convincing. I acknowledge the authors added an additional experiment with the Transformer backbone in the rebuttal. I would request to scale up the training by increasing the model sizes and compute budget until the job can finish within 1-2 weeks. Hopefully with this setting, we can see if the conclusion generalizes to large-scale settings.

---

> ### Author Response · Authors · 2026-01-13
>
> We thank the Action Editor for the constructive feedback. We have submitted the camera-ready version with revisions that address the requested points.
>
> 1. **Extending beyond supervised pretraining (Section 4.6):**
>
> To address the Editor’s request, we added a new subsection (Section 4.6) that evaluates our hypothesis in an unsupervised (self-supervised) pretraining setting. Specifically, we introduce additional experiments using a transformer-based backbone (EquiformerV2) pretrained with the DeNS framework, a recent TMLR work [1] that extends self-supervised denoising to non-equilibrium molecular structures. Denoising pretraining learns representations by training the model to recover the original molecular geometries from artificially perturbed (noisy) atomic positions, and is one of the most widely used self-supervised objectives for 3D molecular graphs.
>
> The results, summarized in Table 4, demonstrate that CSI remains a robust predictor of downstream performance even in the self-supervised setting. Across all tasks, the upstream dataset with the lowest CSI (i.e., ANI-1x) yields the most effective pretraining for downstream tasks. We thank the Editor for this suggestion as it once again highlights clearly that our proposed metric generalizes beyond supervised learning to other learning paradigms, reinforcing the importance of dataset alignment.
>
> [1] Liao, Yi-Lun, et al. "Generalizing Denoising to Non-Equilibrium Structures Improves Equivariant Force Fields." Transactions on Machine Learning Research.
>
> ---
> 2. **Backbone types and model sizes (Section 4.4):**
>
> We added a new subsection (Section 4.4), expanding beyond the preliminary transformer experiment included in the rebuttal, and scaled up the evaluation to two EquiformerV2 model sizes (31M and 87M parameters) trained with a 10M pretraining budget. Pretraining the 87M model at this scale requires up to 17 A100 GPU-days (see Table 3 for full details), which aligns with the Editor’s requested 1–2 week compute scale. These results confirm that our conclusions remain consistent under larger models and higher compute budgets.
>
> ---
> We have also proofread the paper and improved the writing in several parts. Overall, we thank the Editor for these suggestions as we believe these additions significantly strengthened the paper and reinforced our central conclusion: dataset alignment is a fundamental driver of effective pretraining for 3D representation learning.
>
> We are grateful for the Editor’s guidance and are happy to provide any further clarification if needed.